# Accurate, Private, Secure, Federated U-statistics with Higher Degree

**Quentin Sinh** [1]   **Jan Ramon** [1]

## Abstract

We study the problem of computing a U-statistic with a kernel function $f$ of degree $k \geq 2$, i.e., the average of some function $f$ over all $k$-tuples of instances, in a federated learning setting. U-statistics of degree 2 include several useful statistics such as Kendall's $\tau$ coefficient, the Area under the Receiver-Operator Curve and the Gini mean difference. Existing methods provide solutions only under the lower-utility local differential privacy model and/or scale poorly in the size of the domain discretization.

In this work, we propose a protocol that securely computes U-statistics of degree $k \geq 2$ under central differential privacy by leveraging Multi Party Computation (MPC). Our method substantially improves accuracy when compared to prior solutions. We provide a detailed theoretical analysis of its accuracy, communication and computational properties. We evaluate its performance empirically, obtaining favorable results, e.g., for Kendall's $\tau$ coefficient, our approach reduces the Mean Squared Error by up to four orders of magnitude over existing baselines.

## 1. Introduction

In statistical theory, U-statistics are a class of statistics widely used for estimating population parameters. Introduced by W. Hoeffding (Hoeffding, 1992), a U-statistic with kernel of degree $k$ is an estimator returning the average of a kernel function over all possible $k$-tuples of instances. Many statistics can be expressed as a U-statistic. E.g., the sample mean is a U-statistic with a kernel function of degree 1. Kendall's $\tau$, the Gini mean difference and the Area under the ROC curve are U-statistics with a kernel of degree 2. Several problems in machine learning use U-statistics of

[1]MAGNET, Univ. Lille, INRIA, CNRS, UMR 9189 - CRIStAL, F-59000 Lille, France. Correspondence to: Quentin Sinh <quentin.sinh@inria.fr>, Jan Ramon <jan.ramon@inria.fr>.

*Proceedings of the 43rd International Conference on Machine Learning*, Seoul, South Korea. PMLR 306, 2026. Copyright 2026 by the author(s).

degree $k \geq 2$, e.g., supervised metric learning (Bellet et al., 2015) where the goal is to construct a task-specific distance metric from data, pairwise clustering of data (Clémençon, 2011) where one wants to partition the data into homogenous groups, or multi-partite ranking (Clémençon et al., 2013) which aims at ordering data points given some score function.

With the exponential growth in data generation and its usage, data privacy has become a major issue. To protect data, it is important to avoid disclosing intermediate results and outputs from which input data properties can be inferred. One can avoid leaks from intermediate results either by encrypting messages or by adding noise to the information exchanged, while to protect outputs the only option is to make them noisy. Introduced by (Dwork et al., 2006a), Differential Privacy (DP) is a framework that is well-suited for making information noisy and trading off its privacy against its utility.

In cases where data is stored at the premises of multiple data owners, Federated Learning (FL) avoids the need to transfer the data to a central place. Existing FL algorithms commonly compute U-statistics with kernel of degree 1. To avoid leaking information, several strategies exist for secure aggregation, e.g., (Sabater et al., 2022). For instance, in Federated Stochastic Gradient Descent (FedSGD) (Shokri & Shmatikov, 2015), in every step each party computes a local gradient from their local data and the central server computes an average gradient by aggregating over these local gradients. While many existing approaches privately compute U-statistics with a kernel of degree 1 in a satisfactory way, computing privately U-statistics with a kernel of degree $k \geq 2$ received less attention in the literature.

In this paper, we address the problem of computing privately a U-statistic with kernel $f$ of degree $k \geq 2$. We consider parties $P_i$, $i \in [n] = \{1 \ldots n\}$, which each have a single instance $x_i$ and collaboratively compute the U-statistic $U_f = \sum \{ f(x_{v_1} \ldots x_{v_k}) \mid v \subseteq [n] \land |v| = k \} / \binom{n}{k}$ without revealing their input $x_i$. While we assume every party only has one data instance, our approach can be easily generalized as one can simply merge parties together and eliminate the communication cost between them. Before the resulting statistic is released, it is protected by differential privacy. In addition to ensuring privacy, our goal is to de-

sign a procedure that balances two competing objectives: the Mean Squared Error (MSE) between the estimated and true values and the total communication cost required for the computation. These objectives are generally in tension, e.g., reducing communication by sending lower-precision numbers typically increases estimation error. As a secondary objective we aim to minimize the computational cost.

## 1.1. Related works

**Federated setting** In this setting, the aggregator is considered untrusted, implying that individual inputs should remain private. In (Bell et al., 2020), a method is proposed to compute privately an approximation of a U-statistic $\bar{U}_f$ of a kernel function $f$ of degree 2 under $\epsilon$-local differential privacy (LDP) where each party $P_i$ has an input $x_i$ for $i \in [n]$. In LDP, each data owner sends a noisy version of their data $\tilde{x}_i$ to an untrusted aggregator. In this way, the aggregator cannot learn the private data $x_i$. However, in LDP, the quantity of noise added is essentially larger compared to the central DP model, which can affect accuracy of the computed statistic. For $\epsilon$-LDP, under the assumption that $f$ is a Lipschitz function, Bell et al. (Bell et al., 2020) showed that the population MSE of their approximation is bounded by $O(1/\sqrt{n}\epsilon)$. The technique discretizes the input space $\mathbb{X}$ into $t$ bins, and then approximates the kernel $f$ with a matrix $A \in \mathbb{R}^{t \times t}$. The protocol is non-interactive, i.e., parties send only one message to the aggregator.

The authors also propose a second, interactive protocol for privately computing a U-statistic in the 2-party setting (parties $P_1$ and $P_2$ each have a part of the dataset). The protocol samples a subset $\tilde{C}$ containing pairs $(x_i, x_j) \in S_1 \times S_2$ to compute the U-statistic $\bar{U}_{f,\tilde{C}}$ and leverages garbled circuits to compute privately $f(x_i, x_j)$. The resulting U-statistic $\bar{U}_{f,\tilde{C}}$ is said to be incomplete as $\tilde{C} \subset C_2^n$ where $C_2^n$ represents the set of all the possible pairs $(x_i, x_j)$. The MSE is of order $O(\frac{1}{Pn} + \frac{P}{n\epsilon^2})$ where $P$ is linear in the size of the sample $\tilde{C}$. The scheme is $\epsilon$-Computationally DP.

The work of Ghazi et al. (Ghazi et al., 2024) too provides a non-interactive protocol for computing degree 2 U-statistics. In this scheme too, each party $P_i$ has one input instance $x_i \in \mathbb{X}$. Similarly to the non-interactive protocol in (Bell et al., 2020), first, the input space $\mathbb{X}$ is discretized in $t$ bins and the kernel function $f$ is approximated with a matrix $A \in \mathbb{R}^{t \times t}$. In particular, for $i, j \in [t]$, $A_{i,j} = f(r_i, r_j)$ where $r_i, r_j$ are the representative values for bins $i$ and $j$. By leveraging the Johnson-Lindenstrauss (JL) theorem, the matrix $A$ can be approximated by $A \approx L^T R$ where $L, R \in \mathbb{R}^{d \times t}$ for $d = O(\log n)$, enabling a reduction in communication cost. The aggregator publicly releases the matrices $L, R$, after which each party $P_i$ sends $o_i^R = R1_{x_i} + z_i^R$ and $o_i^L = L1_{x_i} + z_i^L$ to the aggregator where $z_i^R$ and $z_i^L$ are DP noise terms. Finally, the aggregator then

computes $\tilde{U}_f = \langle \frac{1}{n}(o_1^L + \cdots + o_n^L), \frac{1}{n}(o_1^R + \cdots + o_n^R)\rangle$. This work provides lower and upper for $\text{MSE}(\tilde{U}_f)$ such that $O\left(\frac{\rho(A,n)^2}{\epsilon^2 n}\right) \leq \text{MSE}(\tilde{U}_f) \leq O\left(\frac{\rho(A,n)^2(\log t)}{\epsilon^2 n}\right)$ where $\epsilon$ is the DP parameter and $\rho(A, n)$ is the minimum of the approximate-factorization norm plus some term proportional to $\sqrt{n}$. Their work also notes that the scheme can be extended to central $(\epsilon, \delta)$-DP through the shuffled model. In this model, each party transmits its noisy data $x_i'$ to a data collector, which relies on a set of $r$ trusted shufflers $\{\mathcal{SH}_j\}_{j \in [r]}$ that anonymize the inputs before sending them to the aggregator. It achieves a MSE $O\left(\frac{\rho(A,n)^2(\log t)}{\epsilon^2 n^2}\right)$.

The authors also propose a three-round algorithm to compute $\tilde{U}_f$ under $\epsilon$-LDP to have a more precise MSE : $O\left(\frac{\|A\|_\infty^2}{\epsilon^2 n}\right)$. It uses a similar technique from the non-interactive algorithm but utilizes clipping and $\epsilon$-local DP mechanisms to estimate some terms in order to calibrate the noise used.

**Central setting** Works explored the intersection of DP and U-statistics within the central setting, i.e., the trusted aggregator has access to all sensitive data belonging to possibly distinct data owners. A notable contribution is the work of (Chaudhuri et al., 2024), who introduced private protocols for two classes: sub-Gaussian U-statistics (where the kernel function's tails decay at least as fast as a Gaussian distribution) and degenerate U-statistics (where the kernel function's first-order projection has zero variance). Their approach yields a mean squared error caused by noise addition of order $\tilde{O}(\frac{k\sqrt{\tau}}{n\epsilon})$ for the sub-Gaussian case and $\tilde{O}(\frac{k^{3/2}C}{n^{3/2}\epsilon})$ for the degenerate case, where $\tau$ represents the sub-Gaussian parameter and $C$ denotes a constant.

**DP on Graph-structured Data** Our work also shares some conceptual links with the DP graph literature, specifically on private conjunctive queries (Narayan & Haeberlen, 2012; Dong & Yi, 2022). While the estimation of U-statistics and graph statistics are not strictly equivalent (representing vertex adjacencies requires non-constant space compared to fixed-size feature vectors), the underlying combinatorial challenges are similar in terms of scaling of the kernel degree $k$ and the sampling of $k$-tuples of parties. Highlighting these links provides a more comprehensive context on the challenges of high-degree U-statistic estimation.

## 1.2. Contributions

In general, existing work either suffers from more noisy output due to the use of LDP or JL approximations, or is limited to a 2-party setting or a security model where trusted parties or shufflers are available.

To address these gaps, in this work we address the problem of computing a U-statistic with a kernel function of degree 2 or more under central differential privacy in the **federated setting** among an arbitrary number of parties.

- We provide the first generic protocol for this task that scales well in the kernel degree $k$, the number of parties $n$ and the size of the discretization in the federated setting. We leverage multi party computation (MPC), keeping computation and communication cost low by making the common assumption that at least a fraction of the parties is honest.

- We present a detailed comparison of our approach against state-of-the-art (SOTA) solutions. Our experiments demonstrate that our protocol achieves improved accuracy and reduced total communication and computation costs compared to Ghazi et al. (Ghazi et al., 2024). Although the solution by Bell et al. (Bell et al., 2020) has similar communication and per-party computation, it suffers from higher MSE and increased server-side computation.

- We further empirically validate our approach on U-statistics with kernel functions of degrees 2 and 3, e.g., the Gini mean difference, Kendall's $\tau$ coefficient, Triangle counting, etc.

The remainder of the article is structured as follows. Sec 2 provides some notations and background. Next, we describe the solution we propose in Sec 3. In Sec 4 we analyze the properties of this protocol and in Sec 5 we compare our protocol with the state of the art. In Sec 6, we present an experimental evaluation. We conclude and offer direction for future work in Sec 7.

## 2. Background

**Notations** We use $[s]$ to denote the set of $s$ smallest positive integers $\{1 \ldots s\}$, $\mathbb{Z}$ to denote the set of all integers and $\mathbb{Q}$ to denote the set of rational numbers. We define $|S|$ to be the cardinality of the set $S$. We denote the indicator function by $\mathbb{I}[\cdot]$, i.e., $\mathbb{I}[true] = 1$ and $\mathbb{I}[false] = 0$. We use $[a, b]$ to represent the set of all real numbers $x$ such that $a \le x \le b$ and $(a, b)$ for the set of all real numbers $x$ such that $a < x < b$. We write $x \oplus y$ to represent the bitwise XOR of integers $x$ and $y$. We define $C_t^s = \binom{[s]}{t}$ to be the set of all unordered tuples (sets) of $t$ distinct elements of $[s]$. From now on, we will simply refer to these unordered tuples as tuples. There holds $|C_t^s| = \binom{s}{t}$. For a sequence $x \in X^s$ of elements of some set $X$ and for a set of indices $v \in [s]^t$, we will denote by $x_v = (x_{v_1}, \ldots, x_{v_t})$ the tuple of $t$ elements obtained by indexing $x$ with the indices in

$v$. For a function $f$ with $t$ arguments, we will also abuse notation to write $f(x_v) = f(x_{v_1}, \ldots, x_{v_t})$.

**Probabilities** We use $\mathbb{P}(A)$ to represent the probability of event $A$. We write $x \sim D$ to express that $x$ is sampled from probability distribution $D$. We consider an instance space $\mathbb{X}$ and a population distribution $P_{\mathbb{X}}$ over $\mathbb{X}$. We assume that every party $i \in [n]$ has a single instance $x_i$ drawn i.i.d. from $P_{\mathbb{X}}$. Let $\mathbb{Y}$ be an output space. Hereafter, let $n$ denote both the number of parties and the number of data points.

**U-statistics** U-statistics are an important concept:

**Definition 2.1** (U-statistic). Let $f : \mathbb{X}^k \to \mathbb{Y}$ be a symmetric function, i.e., for all $x \in \mathbb{X}$ and any permutation $\phi \in S_k$ where $S_k$ denotes the set of all permutations on $[k]$, there holds $f(x_{v_1}, \ldots, x_{v_k}) = f(x_{v_{\phi(1)}}, \ldots, x_{v_{\phi(k)}})$. We call $k$ the degree of $f$. For a set $C \subseteq C_k^n$, we define

$$U_{f,C} = \frac{1}{|C|} \sum_{v \in C_k^n} f(x_v) \qquad (1)$$

We call $U_{f,C_k^n}$ the U-statistic with kernel $f$. For subsets $C \subseteq C_k^n$, we call $U_{f,C}$ a partial U-statistic with kernel $f$.

The symmetry of $f$ ensures that $f(x_v)$, with $v$ a set, is well-defined. Since the cost of evaluating $f(x_v)$ for all $v \in C_k^n$ grows as $O(n^k)$, we (similarly to (Bell et al., 2020)'s interactive 2 party protocol) approximate the U-statistic $U_{f,C_k^n}$ by $U_{f,C}$ for a smaller set $C$. Examples of U-statistics, e.g., Kendall $\tau$, are discussed in Appendix A.5.

**Data Representation** We employ fixed-precision arithmetic, using $l$ bits to represent numbers in $\mathbb{Q}_h = \{\bar{x}^h \in \mathbb{Q} \mid \bar{x}^h = \text{int}(x) \cdot h; x \in \mathbb{F}_{2^\ell}\}$ where $\text{int} : \mathbb{F}_{2^l} \to \{-2^{l-c-1} \ldots 2^{l-c-1} - 1\}$. For details see Appendix A.1

**Secret Sharing** Our protocol relies on secret sharing. In particular, for integers $t \le p \le n$ we consider $(t, p)$-threshold secret sharing. For a secret $x$, we denote by $[\![x]\!]$ a secret sharing of $x$ where $[\![x]\!]_P \in C_p^n$ denotes the indices of parties involved in the sharing of $x$ and for every $i \in [\![x]\!]_P$ the number $[\![x]\!]_i$ is the share of party $P_i$. If $t$ parties in $[\![x]\!]_P$ collaborate, they can reconstruct $x = \text{Rec}([\![x]\!]_{i_1}, \ldots, [\![x]\!]_{i_t})$, while a set of less than $t$ parties are unable to reconstruct the secret. In many applications of secret sharing $[\![x]\!]_P$ is the same for all secret sharings, typically the set of all parties, however our protocol aims to be more efficient. When we consider multiple secret sharings of the same secret $x$ among multiple sets of parties $e$, we use a superscript $[\![x]\!]^{(e)}$ to distinguish them. For more details, see Appendix A.2.

**Preprocessing and evaluation metrics** When the same algorithm is ran repeatedly on different data, we call the *offline phase* the work that is performed once and can be

re-used in every run, while we call the *online phase* the work that is repeated in every run. When evaluating algorithms, we focus on the online phase. With *round complexity* we refer to the number of rounds required in an interactive protocol. With *communication complexity* we refer to the total volume of data exchanged. For more details, see Appendix A.3

**Differential Privacy (DP)**    DP allows for releasing sensitive information by adding some noise. Two datasets are adjacent if they differ in only one instance. For $\epsilon, \delta > 0$, a randomized algorithm $\mathcal{A}$ is said to provide (central) $(\epsilon, \delta)$-DP if for all adjacent datasets $D_1, D_2$ and for all possible subsets $O$ of the range of $\mathcal{A}$ there holds $\mathbb{P}[\mathcal{A}(D_1) \in O] \leq e^\epsilon \cdot \mathbb{P}[\mathcal{A}(D_2) \in O] + \delta$. While central DP only puts a constraint on the output(s), local DP (LPD) requires the input to be already private so that no security is required during the computation. This comes at the cost of lower utility.

For a function $f$, the $q$-sensitivity of $f$ is defined by $\Delta_q f = \max\{\|f(x) - f(y)\|_q \mid \|x - y\| \leq 1\}$, where $q = 2$ if omitted. If $f$ gets a dataset as input, $\|x - y\| \leq 1$ means that $x$ and $y$ are adjacent datasets. Given a value $f(x)$, it can be privatized by applying a DP mechanism, e.g., the Laplace mechanism which adds a value randomly drawn from $Lap(0, \Delta_1 f/\epsilon)$ to $f(x)$ ensures the sum is $\epsilon$-DP, while the Gaussian mechanism which adds a value randomly drawn from $Gauss(0, 2\log(1.25/\delta)(\Delta f)^2/\epsilon^2)$ ensures the sum is $(\epsilon, \delta)$-DP. For more details, see Appendix A.4 or a broad and systematic introduction in (Dwork et al., 2014).

## 3. Proposed protocol

In this section, we present our novel algorithm.

**Threat model**    We assume that there are secure communication channels between parties. We assume an adversary which is static, i.e., a fixed set of parties are corrupted before the start of the protocol, and semi-honest, i.e., the corrupted parties execute correctly the protocol but are willing to cooperate between them to disclose sensitive information. We consider two threat models. In model $\mathcal{M}_{\text{Dis}}$, we assume that the adversary can corrupt $n - 1$ parties. In the model $\mathcal{M}_{\text{HF}}$, we assume that the adversary can corrupt at most $(1 - f_H)(n - 1)$ parties, where $f_H > 0$.

**Partial U-statistic**    The cost of computing $f(x_v)$ over all $v \in C_k^n$ is exponential in $k$. For large datasets, runtimes superlinear in the data are often considered untractable. Therefore, we compute a partial U-statistic $U_{f,E}$ closely approximating $U_{f,C_k^n}$. Here, $E \subseteq C_k^n$ induces a hypergraph $G = (V, E)$ with vertex set $V = [n]$ and edge set $E$. The parties can first jointly generate a seed for a pseudorandom number generator (PRG) and then use it to all draw the same random $E$. For a set $S$, let $E_S$ be the set of edges in $E$ that

contain $S$, i.e., $E_S = \{e \in E : S \subseteq e\}$. We also define the maximal degree $\delta_G^{max} = \max_{i \in [n]} |E_{\{i\}}|$.

**Main protocol $\prod_{\text{U-MPC}}$**    Our protocol uses additive secret sharing, where all parties over whom a secret has been distributed need to collaborate to reconstruct a secret. After an offline phase where data structures such as common randomness are generated, in the online phase which is detailed in Protocol 3.1 the parties first compute a sharing $[\![f(e)]\!]$ for all $e \in E$ by secret-sharing their data $x_i$ (Phase 1) and secret-shared computations (Phase 2). Possibly in parallel, they also compute a sharing $[\![\eta]\!]$ of appropriate DP noise (Phase 3). Finally, they jointly compute the DP sum $[\![\eta + \sum_{e \in E} f(e)]\!]$ and reveal the result (Phase 4). Note that our protocol $\prod_{\text{U-MPC}}$ also supports the case when $k = 1$. In this case, the evaluation of the function $f$ can be done locally.

---

**Protocol $\prod_{\text{U-MPC}}$**

**Input**:
- $G = (V, E)$ with $V = [n]$ and $E \subseteq C_k^n$.
- $\epsilon > 0, \delta > 0$ : DP parameters
- $\Delta f$: the $q$-sensitivity of $f$
- $x = (x_i)_{i \in [n]}$ : the input data

**Online phase**:

1. *Sharing phase*:

   For every $e \in E$ and $j \in [k]$:

   (a) Party $P_{e_j}$ creates secret sharing $[\![x_{e_j}]\!]^{(e)}$ of $x_{e_j}$ among parties $[\![x_{e_j}]\!]_P^{(e)} = e$
   (b) For $l \in [k]$, party $P_{e_j}$ sends $[\![x_{e_j}]\!]^{(e)}$ to $P_{e_l}$.
   (c) $P_{e_j}$ collects $\mathcal{S}_{e_j}^e = \{(e', [\![x_{e'}]\!]_{e_j}^{(e)}) \mid e' \in e\}$

2. *Computing phase*:

   For $e \in E$, $j \in [k]$, party $P_{e_j}$ calls functionality $\mathcal{F}_f(e, e_j, \mathcal{S}_{e_j}^e)$ getting a share $[\![f(x_e)]\!]_{e_j}$.

3. *Noise generation phase* :

   For $i \in [n]$, party $P_i$ calls functionality $\mathcal{F}_{\text{noise}}(\epsilon, \delta, \delta_G^{max}\Delta f)$ receiving a share $[\![\eta]\!]_i$.

4. *Aggregation phase*:

   (a) For $i \in [n]$, party $P_i$ computes $z_i = [\![\eta]\!]_i + \sum_{e \in E_{\{i\}}} [\![f(x_e)]\!]_i$.
   (b) For $i \in [n]$, $P_i$ sends $z_i$ to the aggregator, who reveals $\hat{U}_{f,E} = \frac{1}{|E|} \cdot \text{Rec}(\{z_i\}_{i \in [n]})$.

---

*Protocol 3.1.* Protocol $\prod_{\text{U-MPC}}$ for computing U-statistic with kernel function $f$ of degree $k \geq 2$.

**Functionalities** The main protocol calls two functionalities, which are performed using secret shared computations:

- $\mathcal{F}_f$ evaluates $f(x_e)$
- $\mathcal{F}_{\text{noise}}$ draws random noise $\eta$

Such a functionality can be securely realized by protocols like the GMW protocol (Goldreich et al., 2019) for honest majorities or additive secret sharing protocols for more malicious settings. For $\mathcal{F}_{\text{noise}}$, protocols such as (Eigner et al., 2014; Keller et al., 2024; Sabater et al., 2023) have been proposed. Our protocol make black-box use of functionalities $\mathcal{F}_f$ and $\mathcal{F}_{\text{noise}}$. In this way, our protocol can benefit from any advancements in protocols for the generation of shared noise or in the secure evaluation of a function $f$. Some possible implementations for $\mathcal{F}_{\text{noise}}$ are presented in Appendix B.

# 4. Properties

We now outline the properties of our protocol: correctness, security, privacy, costs and utility. To enable comparison with benchmark solutions, which consider only the case $k = 2$ under $\epsilon$-differential privacy, we derive a proposition summarizing the results for this specific setting.

**Correctness** It is easy to see that our protocol correctly computes $U_{f,E} + \eta/|E|$. For this, the most important step is to observe that due to the use of additive secret sharing the mixing of secret shares over different groups of parties in step 4(a) is sound. Appendix C.1 gives more details.

**Privacy and security** We prove in Appendix C.2 that our algorithm is secure and $(\epsilon, \delta)$-DP. The main observation to show privacy is the sensitivity computation: $U_{f,E}$ is an average of $|E|$ terms of which only at most $\delta_G^{max}$ are affected by the change of a single instance.

**Communication complexity** In Appendix C.3, we show that the protocol requires $O(|E|(k^2\ell + \mathsf{C}_f^C) + \mathsf{C}_\eta^C)$ bits of communication in $\max(\mathsf{C}_f^R, \mathsf{C}_\eta^R) + 2$ rounds where $\mathsf{C}_f^C$ and $\mathsf{C}_\eta^C$ are the communication costs of $\mathcal{F}_f$ and $\mathcal{F}_{\text{noise}}$ respectively, and $\mathsf{C}_f^R$ and $\mathsf{C}_\eta^R$ are the number of rounds needed for $\mathcal{F}_f$ and $\mathcal{F}_{\text{noise}}$ respectively.

The cost which is potentially most expensive from an asymptotic point of view is the communication cost $\mathsf{C}_\eta^C$ of $\mathcal{F}_{\text{noise}}$, as secret shared computation has in general a total communication cost quadratic in the number of parties $n$. However, one can mitigate this problem if one is willing to accept a slightly weaker threat model. In particular, consider the model $\mathcal{M}_{\text{HF}}$ where the adversary corrupts at most a fraction $(1 - f_H)$ of the parties. If we would delegate the computation of $\eta$ to a subgroup $Z \subseteq [n]$ of parties, then the probability that all members of that subgroup $Z$

are dishonest is bounded by $(1 - f_H)^{|Z|}$. Hence, for any negligible probability $\varepsilon$, a randomly selected group $Z$ of $\lceil \log(\varepsilon)/\log((1 - f_H)) \rceil$ will have at least one honest party with probability $1 - \varepsilon$. If we let this group $Z$ privately draw $\eta$ and then secret-share it with all parties, the communication cost becomes linear in $n$.

**Utility** An important question concerns the quality of the approximation made by an approach. We are therefore interested in the mean squared error (MSE) between the population statistic $U_f$ and the output of the protocol $\hat{U}_{f,E} = U_{f,E} + \eta/|E|$. In Appendix D.1.2 we write the MSE $E_{tot} = U_f - \hat{U}_{f,E}$ as a sum $E_{tot} = E_{sample} + E_{inc} + E_{DP}$ where $E_{sample}$ reflects the error of sampling $n$ instances, $E_{inc}$ reflects the error of sampling $E$ and $E_{DP}$ reflects the error due to the DP noise. We show that

$$
\begin{aligned}
E_{sample} &\leq 4/n \\
E_{inc} &\leq \left( \binom{n}{k} - |E| \right) \Big/ 4|E| \left( \binom{n}{k} - 1 \right) \\
E_{DP} &\leq 2k^2(\Delta f)^2/n^2\epsilon^2
\end{aligned}
$$

These bounds assume that $E$ is sampled in a way which makes the graph $G$ as regular as possible, so all vertices have degree $\lceil |E|k/n \rceil$ or $\lfloor |E|k/n \rfloor$. Appendix F presents a simple method to draw such $E$.

# 5. Comparison

In this section, we compare the several approaches under different security settings, using various metrics for the computation a U-statistic of degree $k = 2$ under $\epsilon$-DP:

- Ghazi: non-interactive protocol in (Ghazi et al., 2024)
- GhaziSM: the non-interactive protocol from (Ghazi et al., 2024) in the shuffled model, with $r$ servers to shuffle/anonymize messages, following the implementation of (Balle et al., 2020, Sect. 5),
- Bell: the generic LDP protocol in (Bell et al., 2020)
- our protocol $\prod_{\text{U-MPC}}$ under BalancedSamp (Algorithm 1) for sampling the edges, either under $\mathcal{M}_{\text{Dis}}$ with $O(n^2)$ noise generation cost or under $\mathcal{M}_{\text{HF}}$ with $O(n)$ noise generation cost.

This comparison doesn't consider (Bell et al., 2020)'s generatic protocol from 2PC as it is limited to 2 parties.

We summarize the comparison results in Table 1. Expressions in Table 1 omit terms which are not asymptotically dominant.

The number of bits required to represent one element is denoted by $\ell$. The notation $\mathsf{c}_\eta^C$ is used for a constant factor in the communication cost of securely drawing noise independent of $n$. The term $\mathsf{C}_f^C$ represents the communication cost incurred during the online phase of the evaluation of $f$

in $\prod_{\text{U-MPC}}$. while $\mathsf{C}_f^T$ represents the computational cost per party incurred during the online phase of the evaluation of $f$ in Protocol $\prod_{\text{U-MPC}}$.

**Mean squared error**  Derivations of the MSE in Table 1 can be found in Appendix D.1. We omit the sampling error $E_{sample}$ from our analysis as all compared methods start from a sample and suffer a similar error. One can observe that the protocols of Bell and Ghazi don't offer good MSE for fine-grained discretizations (parameter $t$), while in our protocol the discretization is only relevant to represent values as integers and doesn't negatively affect the MSE.

**Cost**  The table presents two communication costs of our protocol depending on the threat model considered. For $\mathcal{M}_{\text{HF}}$, our communication cost asymptotically matches the best baseline protocol Bell while it outperforms Bell on other metrics. concerning the computation cost, one can observe that in contrast to baseline protocols, in our protocol the computation cost per party does not depend more than logarithmically on the discretization size $t$. More details are provided in Appendix D.2.

**Scalability**  Computing a U-statistic of kernel degree $k > 2$ is typically hindered by a combinatorial explosion, as the number of tuples grows as $O(n^k)$. To address this intractability, our protocol maintains linear communication cost by restricting the size of the set of sampled edges to $|E| = O(n)$. This configuration achieves an optimal utility-communication trade-off: since the sampling error $E_{sample} = O(1/n)$ and the error $E_{inc}$ scales as $O(1/|E|)$, increasing the sample size beyond $O(n)$ can't result in a total error with order of magnitude smaller than $O(1/n)$. To our knowledge, no existing SOTA protocol provides a privacy-preserving solution for computing a U-statistic of degree $k > 2$.

# 6. Experiments

In this section, we present an empirical evaluation, limited to degree-2 U-statistics for the purpose of comparability with existing baselines.

## 6.1. Experimental setup

**Questions**  We consider the following experimental questions:

Q1 How does our protocol compare in terms of online communication cost, online computational cost and MSE against the baselines ?

Q2 How does our sampling algorithm BalancedSamp (Algorithm 1) reduce the MSE compared to other sampling methods ?

Q3 How does our protocol perform on a kernel function $k > 2$ ?

**Protocols**  When comparing with existing approaches, we consider the 4 protocols Ghazi, GhaziSM, Bell and Umpc under the threat model $\mathcal{M}_{\text{HF}}$ defined in the beginning of Sec 5.

**Evaluation Metrics**  We measure the communication cost as the total number of bits exchanged between the parties during the online phase of the protocol. This metric provides a more stable evaluation than empirical runtimes, which are often influenced by hard to control external factors such as the load of the network, the network latency, etc. Similarly, we measure for computation cost through the total number of operations performed by a single party, as this remains invariant to environmental factors such as OS or hardware-specific scheduling. Finally, we report the MSE along with the Standard Error of Mean (SEM).

**Datasets**  We perform our experiments using the following datasets.

- *Synthetic dataset*: We consider a synthetic dataset where $\mathbb{X} = [0, 1]$ and $x_i \sim \text{Uni}([0, 1])$ for $i \in [n]$ where $\text{Uni}(S)$ is the uniform distribution over the set $S$. We use this synthetic data to measure the effect of variation in dataset size on the relevant metrics.

- *Bank Marketing dataset* (Moro & Cortez, 2014): This dataset is related to marketing campaigns of a Portuguese banking institution. The classification goal is to predict if the client will subscribe to a term deposit. It contains 16 features for each client. The dataset contains 4521 instances. We normalize all numerical dataset values such that $\mathbb{X} = [0, 1]$.

- *Social circles: Facebook dataset* (Leskovec & Mcauley, 2012): This dataset consists of an undirect graph representing friendship netwroks from Facebook. It comprises 4039 nodes and 88234 edges.

We present further empirical evaluations on additional datasets in Appendix E.

**Parameters**  As discussed in Sec 2 every party $P_i$ with $i \in [n]$ holds a value $x_i \in \mathbb{X}$ which we represent using a fixed-precision representation. We use integers of $\ell = 40$ bits with $c = 14$, i.e., with $h = 2^{-14}$. This enables representing the U-statistic with values in the range $[-2^{25}, 2^{25}]$, which is sufficient for our purposes. Elements of the output space $\mathbb{Y}$ are encoded in the same manner. We focus on $\epsilon$-LDP/DP privacy, as it is also considered in the existing solutions. A detailed description of the MPC protocols used can be found in Appendix E.1.

*Table 1.* Protocol comparison

| Protocol | MSE[(1)] | Comm. cost[(2)] | Party comp. cost[(3)] | Server comp. cost[(4)] |
|---|---|---|---|---|
| Bell (Bell et al., 2020) | $\frac{1}{t^2} + \frac{t^2}{n\epsilon^2}$ | $n\ell t$ | $t$ | $n^2 t^2$ |
| Ghazi (Ghazi et al., 2024) | $\frac{1}{t^2} + \frac{t^2 \log t}{\epsilon^2 n}$ | $n^2 \epsilon^2 (\log t)\ell$ | $\epsilon^2 nt \log t$ | $\epsilon^2 n^2 \log t$ |
| GhaziSM (Ghazi et al., 2024) | $\frac{1}{t^2} + \frac{t^2 \log t}{\epsilon^2 n^2}$ | $(\log n)n^2 \epsilon^2 (\log t)\ell$ | $\epsilon^2 nt \log t + t \log n$ | $(\log n)\epsilon^2 n^2 \log t$ |
| Umpc (Prot. 3.1) | $\frac{1}{|E|} + \frac{1}{n^2\epsilon^2}$ | $\mathcal{M}_{\text{Dis}} : \|E\|(\ell + \mathsf{C}_f^C) + n^2\ell\mathsf{c}_\eta^C$ $\mathcal{M}_{\text{HF}} : \|E\|(\ell + \mathsf{C}_f^C) + n\ell\mathsf{c}_\eta^C$ | $\frac{\|E\|}{n} \cdot \mathsf{C}_f^T + n$ | $n$ |

Asymptotic expressions are provided for the MSE $E_{tot} - E_{sample}$ [(1)] , the total communication cost measured in total bits exchanged in the online phase [(2)] and the computation cost per party [(3)] and for the server [(4)].

**Hardware and implementation**   We conduct experiments on a Linux server equipped with a 2.20GHz Intel Xeon processor, 64GB of RAM, and a Tesla P100 GPU (12GB capacity). The experiments compute the MSE, the communication cost and the theoretical computation cost according to our complexity analysis. The code for the experiments is available at `https://github.com/federated-ustat-project/federated-U-statistics.git`.

## 6.2. Experimental results

A complete explanation of the considered U-statistics is available in Appendix A.5. For more experiments, see Appendix E.2.

### 6.2.1. Gini Mean Difference

We use the synthetic dataset to compute the U-statistic $U_f$ where $f(x_i, y_i) = |x_i - y_i|$. We aim to compare the MSE and the total communication cost of the different protocols.

**Results**   Figure 1 shows on the left the online communication cost as a function of the number of parties (and hence dataset size) $n$, and on the right the MSE as a function of $n$, obtained when computing the Gini mean difference across the different protocols.

Figure 2 shows the per-party and server computation costs for $\epsilon = 1$ and $t = 256$.

**Observations**   We observe that our protocol yields a better MSE compared to the alternatives. Out protocol also has a low communication cost, similarly to Bell.

We can see that Bell achieves the lowest per-party computa-

tion cost, at the expense of higher server-side computation. Even so, in this application, the communication cost is likely to be a more important consideration as nowadays computational power is more readily available than scalable communication.

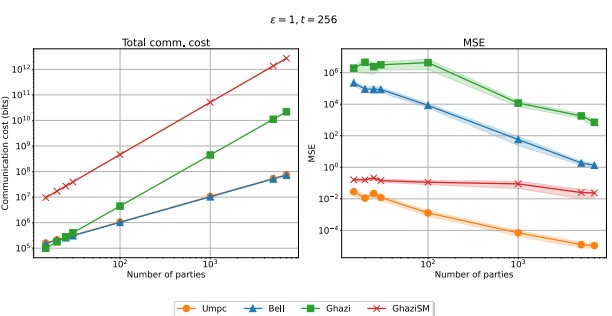

*Figure 1.* Online total communication cost (left) and MSE (right) as a function of the number of parties $n$. for the computation of Gini mean difference for $\epsilon = 1$. The number of discretization bins is set to $t = 256$. Each data point $x_i$ is uniformly drawn from $[0, 1]$. For Umpc, we sample $2n$ edges for $|E|$.

### 6.2.2. Kendall's $\tau$ coefficient

We use the Banking Marketing dataset (Moro & Cortez, 2014). Let $x_i = (y_i, z_i)$ be a data point where $y_i, z_i$ represents respectively the age and the average yearly balance for party $P_i$. We want to compute Kendall's $\tau$ coefficient for those two variables.

**Results**   Figure 3 presents the online total communication cost and the online total computation cost (defined as the sum of the server computation and $n$ times the per-party computation cost) of the different protocols for the computation of the Kendall's $\tau$ coefficient for different values

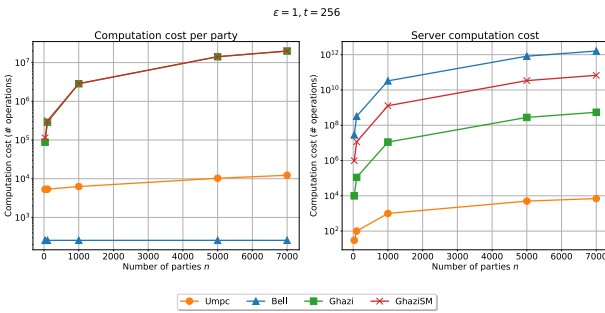
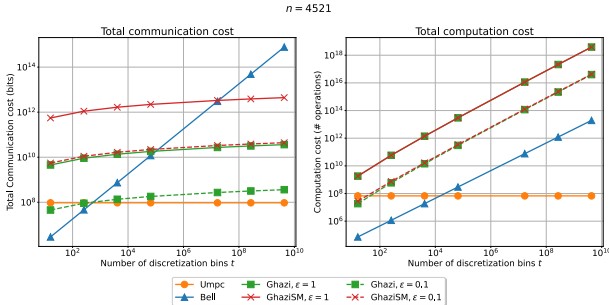

*Figure 2.* Online per-party computation cost (left) and online server computation cost (right) as a function of the number of parties, for the computation of Gini mean difference for $\epsilon = 1$. The number of discretization bins is $t = 256$. Each data point $x_i$ is uniformly drawn from $[0, 1]$. For Umpc, we sample $2n$ pairs for $|E|$.

*Figure 3.* Online total communication cost (left) and online total computation cost (right) for computing the Kendall's $\tau$ coefficient over the number of discretization bins $t$. The total computation cost is defined as the sum of the server computation and $n$ times the per-party computation cost. The communication and computation costs vary with $\epsilon = \{0, 1\}$ only for Ghazi and GhaziSM. The dataset is taken from (Moro & Cortez, 2014) and contains $n = 4521$ entries. For Umpc, we sample $2n$ pairs for $|E|$, i.e., $|E| = 9042$.

of $t$. Since only the costs of Ghazi and GhaziSM depend on the DP parameter $\epsilon$, we plot their communication and computational costs for $\epsilon = \{0.1, 1\}$.

Figure 4 displays the MSE over the number of discretization bins $t$ and the MSE over the online communication cost for $\epsilon = \{0.1, 1\}$. We observe that the protocol Bell requires the lowest communication overhead while our protocol Umpc achieves the lowest MSE.

**Observations** Observe that as the DP parameter $\epsilon$ increases, the protocols Ghazi and GhaziSM require more communication. For example, $\epsilon \geq 1$, Ghazi incurs a higher communication cost than Umpc.

For computational cost, our protocol Umpc requires to compute the kernel function $f(x_i, x_j) = \text{sign}(y_i - y_j)\text{sign}(z_i - z_j)$, which amounts to evaluate 2 comparisons and one product, which can be performed in $O(\ell)$ operations using Function Secret Sharing (FSS) (Boyle et al., 2019), This results in the lowest overall computational cost among all the methods considered when exceeding a certain $t$.

### 6.2.3. DUPLICATE PAIR RATIO

We focus on computing $U_f$ where $f(x, y) = \mathbb{I}[x = y]$ such that $x, y$ are categorical data points. We use the same dataset as in the previous experiment, i.e., the Bank Marketing dataset (Moro & Cortez, 2014). Each data point $x_i$ represents the job of party $P_i$ for $i \in [n]$. In this context, the input space is $\mathbb{X} = \{0, \dots, 11\}$. We seek to analyze how BalancedSamp influences the MSE in comparison to Bernoulli sampling and uniform sampling without replacement, as well as how the MSE scales with the size of $E$.

**Results** Table 2 reports the MSE of the protocol Umpc under various sampling strategies. For this evaluation, we

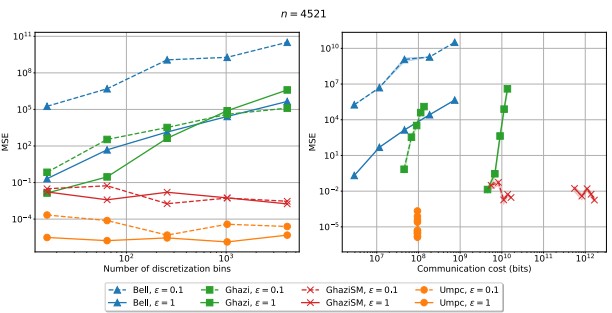

*Figure 4.* MSE over the number of discretization bins $t$ (left) and MSE over the online communication cost (right) for computing the Kendall's $\tau$ coefficient for $\epsilon = \{0.1, 1\}$. The dataset is taken from (Moro & Cortez, 2014) and contains $n = 4521$ entries. For Umpc, we sample $2n$ pairs for $|E|$, i.e., $|E| = 9042$.

compute the duplicate pair ratio over the 'job' attribute from the *Bank Marketing dataset*, setting the privacy budget to $\epsilon = 1$ and the number of edges $|E| = 0.2 \cdot \binom{n}{2}$.

**Observations** We can observe that the balanced sampling strategy outperforms the alternative strategies, achieving a reduction in MSE by factors of approximately $2.5\times$ and $1.75\times$ compared to Bernoulli and uniform sampling without replacement. Furthermore, balanced sampling exhibits greater stability, as shown by a lower SEM.

### 6.2.4. TRIANGLE COUNTING

To demonstrate the support of our protocol Umpc for higher-order U-statistics ($k > 2$), we compute the number of triangles $T_G$ in the graph $G = (W, F)$ derived from the *Social Circle* dataset. The kernel function induced is $f(x, y, z) = \mathbb{I}[(x, y) \in F] \cdot \mathbb{I}[(y, z) \in F] \cdot \mathbb{I}[(x, z) \in F]$. A detailed derivation of the triangle counting formula is

*Table 2.* MSE comparison for protocol Umpc when $E$ is sampled using BalancedSamp (Algorithm 1), Bernoulli sampling and sampling without replacement for computing the duplicate pair ratio for the size of the edge set $|E| = 0.2 \cdot \binom{n}{2}$ with privacy budget $\epsilon = 1$. We use the *Bank Marketing dataset* containing $n = 4521$ data points. MSE is reported as Mean $\pm$ SEM.

| Sampling strategy | MSE |
|---|---|
| Balanced | $2.18 \times 10^{-7} \pm 7.60 \times 10^{-8}$ |
| Uniform | $3.82 \times 10^{-7} \pm 1.34 \times 10^{-7}$ |
| Bernoulli | $5.50 \times 10^{-7} \pm 2.72 \times 10^{-7}$ |

provided in Appendix A.5.

**Results**   Table 3 presents the relative error and the total online communication cost and for our protocol Umpc. The error is expressed as the Root Mean Squared Error (RMSE) divided by the ground truth number of triangles $T_G$. Results are reported for the *Social Circles: Facebook dataset* under balanced sampling with privacy budget $\epsilon = 1$.

**Observations**   In the $|E| = 10^7$ regime, our protocol achieves high utility with a relative error of only $2\%$. The associated online communication overhead remains feasible, totaling $19.4$ GB. This decomposes to approximately $4.8$ MB per party, demonstrating the practical scalability of the Umpc protocol for higher-degree kernel functions ($k = 3$).

*Table 3.* Relative error and total online communication cost for the protocol Umpc using balanced sampling for the triangle count with $\epsilon = 1$ for the *Social Circles: Facebook dataset*. MSE is reported as Mean $\pm$ SEM.

| Size $E$ | RMSE/$T_G$ | Comm. cost |
|---|---|---|
| $10^5$ | $0.25574 \pm 0.03116$ | $194$ MB |
| $10^6$ | $0.07022 \pm 0.00853$ | $1.94$ GB |
| $10^7$ | $0.02375 \pm 0.00390$ | $19.4$ GB |

### 6.3. Key takeaways

In summary, we reply to the experimental questions:

Q1  Umpc results in the lowest MSE and has competitive communication cost. Bell requires the lowest per-party computation cost while our protocol requires less per-party computation than Ghazi et al.'s solution. Regarding overall computation cost, our protocol offers a more efficient alternative to existing methods.

Q2  As evidenced by the lower MSE in Table 2, BalancedSamp (Algorithm 1) provides a more effective sampling strategy, resulting in higher utility than alternative approaches.

Q3  The triangle count experiment demonstrates that the Umpc protocol scales efficiently to higher-degree kernel functions. Specifically, it maintains consistent

utility while requiring only a linear number of edges $|E| = O(n)$.

## 7. Conclusion

In this work, we propose the protocol $\prod_{\text{U-MPC}}$ to compute privately U-statistics. Our contribution is two-fold: (1) We propose the first generic privacy-preserving approach that leverages MPC techniques to compute U-statistics of arbitrary degree $k \geq 2$. (2) For the specific case of U-statistic of degree 2, we reduce the error rate to $O(1/n^2\epsilon^2)$ under central $\epsilon$-DP in a federated setting, achieving lower error than existing solutions while significantly reducing per-party and server-side computation costs and total communication compared to Ghazi et al. (Ghazi et al., 2024). While Bell et al.'s solution (Bell et al., 2020) requires less communication and per-party computation, it incurs higher MSE and server-side computation. We validate these theoretical gains through experimental evaluation.

There are several future lines of work. As not a single strategy is optimal for all criteria, it would be interesting to investigate whether there is a strategy that can for every application select the best option. Second, further improving MPC protocols and the use we make of them could decrease the communication and computation cost.

## Acknowledgments

The authors would like to thank the anonymous ICML 2026 reviewers for their useful suggestions. This work has been funded by the Horizon Europe FLUTE project grant no. 101095382, by the ANR PMR project, and by the French State aid under the France 2030 program with the reference ANR-23-PEIA-005 (REDEEM project).

## Impact Statement

This paper presents work whose goal is to advance the field of Machine Learning. There are many potential societal consequences of our work, the only one we feel important to highlight here is the positive impact of privacy-preserving machine learning which helps to protect the personal data of while their aggregated statistical properties are of great interest.

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

# A. Additional preliminaries

## A.1. Data representation

We employ fixed-precision arithmetic to represent fractional numbers with a fixed number of decimals. To represent an element using fixed-precision arithmetic, we use an element from the finite field $\mathbb{F}_{2^\ell}$ with $\ell \in \mathbb{N}$. Recall that $\mathbb{F}_{2^\ell} = \mathbb{Z}_2[X]/(P)$ where $\mathbb{Z}_2[X]$ is the ring of polynomials modulo 2 and $P$ is an irreducible polynomial of degree $\ell$. We define the mapping $\mathsf{int} : \mathbb{F}_{2^\ell} \to \{0, \ldots, 2^\ell - 1\}$ that assigns to each polynomial $Q$ its integer representation obtained by evaluating $Q$ at $X = 2$. Let $h = 2^{-c}$ be a fixed scaling factor such that $c \leq \ell$. We define $\mathbb{Q}_h = \{\bar{x}^h \in \mathbb{Q} \mid \bar{x}^h = \mathsf{int}(x) \cdot h; x \in \mathbb{F}_{2^\ell}\}$ to be the set of rational numbers we can express where $\cdot$ is the multiplication in $\mathbb{Q}$. We define the function $\mathsf{convert} : \mathbb{Q}_h \to \mathbb{F}_{2^\ell}$ such that $\mathsf{convert}(\bar{a}^h) = \mathsf{int}^{-1}(\bar{a}^h/h) = a$ where $\bar{a}^h \in \mathbb{Q}_h$. We say that $a$ is the element from $\mathbb{F}_{2^\ell}$ representing $\bar{a}^h$.

The sign of a value is contained in the most significant bit of its representation. For any $x \in \mathbb{F}_{2^\ell}$, let $(b_{\ell-1}, b_{\ell-2}, \ldots, b_0)$ be the binary decomposition of $x$. Then, $\bar{x}^h = (2 \cdot b_\ell - 1) \cdot \sum_{i=-b}^{\ell-2} 2^i$. The set $\{0, \ldots, 2^{\ell-1} - 1\}$ represents the set of positive integers while the set $\{2^{\ell-1}, \ldots, 2^\ell - 1\}$ represents the negative integers.

## A.2. Secret sharing

Our protocol relies on MPC and more precisely on secret sharing. We now describe secret sharing (SS). A secret shared value $x$ is denoted $[\![x]\!]$. We will denote by $[\![x]\!]_P \in C_p^n$ the set such that for $i \in [\![x]\!]_P$, party $P_i$ gets a share of $x$. We the share of party $P_i$ by $[\![x]\!]_i$. In many applications of secret sharing, $[\![x]\!]_P$ is the same for all secret sharings, typically the set of all parties, however in the protocol we propose this is not necessarily the case.

**Definition A.1** (Secret Sharing). A $(t, p)$-threshold (information-theoretic secure) Secret Sharing (SS) over a finite field $\mathbb{F}$ allows for sharing a secret $x \in \mathbb{F}$ into $p$ shares such that knowing at least $t$ shares is sufficient for reconstructing the secret. Secret Sharing consists of two algorithms (Share, Rec) with the following syntax:

- Share$(t, p, x)$: given the threshold $t$, the number of parties $p$ and $x \in \mathbb{F}$, outputs shares $[\![x]\!]_1, \ldots, [\![x]\!]_p \in \mathbb{F}$. This algorithm runs in polynomial time and is probabilistic.

- Rec$([\![x]\!]_{i_1}, \ldots, [\![x]\!]_{i_{t'}})$: on input $[\![x]\!]_{i_1}, \ldots, [\![x]\!]_{i_{t'}}$ where $\{i_1, \ldots, i_{t'}\}$ is a subset of $C_{t'}^p$ of size $t \leq t' \leq p$, outputs the underlying secret $x \in \mathbb{F}$. This algorithm runs in polynomial time and is deterministic.

The algorithms (Share, Rec) should satisfy the following correctness and security properties:

- **Correctness**: For every input $x \in \mathbb{F}$ and $t \leq t' \leq p$, we have:

$$\mathbb{P}\big[[\![x]\!]_1, \ldots, [\![x]\!]_p \leftarrow \mathsf{Share}(t, p, x);$$
$$\forall \{i_1, \ldots, i_{t'}\} \subseteq C_{t'}^p, \mathsf{Rec}([\![x]\!]_{i_1}, \ldots, [\![x]\!]_{i_{t'}}) = x\big] = 1. \tag{2}$$

- **Security**: For every input $x \in \mathbb{F}$, let $[\![x]\!]_1, \ldots, [\![x]\!]_p \leftarrow \mathsf{Share}(t, p, x)$. Then, for any set $\{i_1, \ldots, i_{t'}\}$ of size $t' < t$, any set of shares $[\![x]\!]_{i_1}, \ldots, [\![x]\!]_{i_{t'}}$ cannot learn any information on the secret $x$.

In our scheme, we consider additive secret sharing over $(k, k)$-threshold and $(n, n)$-threshold as it it linear and helps us reduce the communication cost.

**Additive secret sharing (AddSS)**  Additive secret sharing is a $(p, p)$-threshold secret sharing. To share a secret $x \in \mathbb{Z}_M$, the algorithm Share$(p, p, x)$ outputs the values $[\![x]\!]_1, \ldots, [\![x]\!]_p$ where $x \equiv \sum_{i \in [p]} [\![x]\!]_p \mod M$. To reconstruct a secret, the algorithm Rec$([\![x]\!]_1, \ldots, [\![x]\!]_p)$ outputs $\sum_{i \in [p]} [\![x]\!]_p \mod M$.

**Other MPC primitives**  While the MPC landscape offers various primitives such as Homomorphic Encryption (HE), we employ AddSS based on three technical reasons. First, HE introduces a computational overhead that is orders of magnitude higher than AddSS. Recent HE schemes, e.g., (Brakerski, 2012; Cheon et al., 2017), are based on Ring-LWE (RLWE). A ciphertext in RLWE consists of two large polynomials from the ring $R_q = \mathbb{Z}_q[X]/(X^N + 1)$, where the degree $N$ and modulus $q$ are parameters scaled with the security parameter $\lambda$. For instance, achieving a standard security level of $\lambda = 128$ with $N = 1024$ requires a modulus of approximatively $q \approx 2^{28}$ (see (Bossuat et al., 2024)). This results in a massive

ciphertext expansion compared to AddSS, which can operate on much smaller moduli. Second, it facilitates the "free" injection of distributed DP noise. We exploit the fact that additive shares are indistinguishable from uniform. Because the convolution of a uniform distribution with any independent random variable remains uniform, privacy can be ensured through local pertubations alone (see remark B.1). Finally, our architecture utilizes AddSS for kernel evaluations that seamlessly generalize to a global sharing scheme across all parties. During the *Computing phase*, parties engage in 2-party MPC with different parties. By adding those shares with the $(n, n)$ secret-shared DP noise, the result becomes an $(n, n)$ secret-shared across the entire network at zero additional communication cost. Those are unique advantages of AddSS that cannot be easily replicated with Shamir secret sharing or HE.

### A.3. MPC Preprocessing and communication

Often, algorithms in general and MPC protocols in particular rely on preprocessing to minimize the amount of work performed or data exchanged by re-using the same preprocessing result in multiple runs of the main algorithm or the main loop.

Such algorithms are divided into two phases:

1. offline phase (executed once): parties generate data structures that can be used repeatedly later on. For example:
    - In MPC algorithms relying on randomness, correlated randomness that is independent of the private inputs.
    - In (Ghazi et al., 2024) and (Bell et al., 2020), the matrix $A \in \mathbb{R}^{t \times t}$ where for $i, j \in [t]$, $A_{i,j} = f(r_i, r_j)$ where $r_i, r_j$ are the representative values for bins $i$ and $j$.

2. online phase (ran for each input): parties use the precomputed data structures to accelerate the actual computation.

In our setting, the online phase corresponds to computing a U-statistic $U_f$. As the offline cost becomes negligible when many such computations are performed, we focus on the online communication overhead of a single computation of $U_f$.

We evaluate communication using two different metrics:

- round complexity: the number of sequential steps in which parties exchange messages that is needed to complete the protocol,

- communication complexity: the total volume of data exchanged during the protocol. It can be expressed as either the number of times a party sends a share to others or as the number of bits exchanged during the protocol. Here, we adopt the latter (bits exchanged) as this eases the comparison with other approaches.

### A.4. Differential Privacy (DP)

Differential Privacy allows for releasing information about a dataset without compromising data of individuals by making the information noisy. There is a trade-off between added noise (and hence privacy) and utility.

We say datasets are adjacent if they differ on at most one instance. There are multiple notions of adjacency. For example, under *replace adjacency*, we say $D_1$ and $D_2$ are adjacent if there is a dataset $D_0$ and instances $x_1, x_2$ such that $D_1 = D_0 \cup \{x_1\}$ and $D_2 = D_0 \cup \{x_2\}$. Under *add adjacency*, we say $D_1$ and $D_2$ are adjacent if there is an instance $x$ such that $D_1 = D_2 \cup \{x\}$ or $D_2 = D_1 \cup \{x\}$.

**Definition A.2** (Central DP). For $\epsilon > 0$, $\delta > 0$ and an adjacency relation, a randomized algorithm $\mathcal{A}$ is said to provide $(\epsilon, \delta)$-differential privacy (DP) if, for all datasets $D_1, D_2$ that are adjacent and for all possible subsets $O$ of the range of $\mathcal{A}$,

$$\mathbb{P}[\mathcal{A}(D_1) \in O] \leq e^\epsilon \cdot \mathbb{P}[\mathcal{A}(D_2) \in O] + \delta. \tag{3}$$

Now, we can define local DP, which, in practice, applies local noise to every instance individually.

**Definition A.3** (Local DP). A local randomized algorithm $\mathcal{A}$ is said to provide $(\epsilon, \delta)$-local differential privacy (LDP) if for all $x, x' \in \mathbb{X}$ and all possible outputs $O$ in the range of $\mathcal{A}$,

$$\mathbb{P}[\mathcal{A}(x) = O] \leq e^\epsilon \cdot \mathbb{P}[\mathcal{A}(x') = O] + \delta. \tag{4}$$

If $f : \mathbb{X}^* \to \mathbb{Y}$ is a function and $A : \mathbb{X} \to \mathbb{X}$ is LDP, then

$$f(\{A(x) \mid x \in D\})$$

is an LDP application of $f$ on the dataset $D \in \mathbb{X}^*$.

We also require the notion of sensitivity which measures the amount of change in the result of a query when adding or removing one's personal data.

**Definition A.4** (Sensitivity). Let $\mathcal{D}$ be a collection of datasets. Let $f : \mathcal{D} \to \mathbb{R}$. Let $d(x, x')$ represent the distance between dataset $x$ and $x'$ where $|\cdot|$ is the cardinality operator, i.e., $d(x, x') = |x \cap x'|$. E.g., $d(x, x') = 1$ means that $x$ and $x'$ differ in at most one instance. The $\ell_1$-sensitivity $\Delta f$ of a function $f$ is defined as:

$$\Delta f = \max_{\substack{x,y \in \mathcal{D} \\ d(x,y) \leq 1}} |f(x) - f(y)|. \tag{5}$$

In a similar way, the $\ell_2$-sensitivity is defined as:

$$\Delta_2 f = \max_{\substack{x,y \in \mathcal{D} \\ d(x,y) \leq 1}} \| f(x) - f(y) \|_2 \tag{6}$$

where $\| \cdot \|_2$ corresponds to the Euclidean norm.

To achieve $(\epsilon, \delta)$-DP, one can add controlled noise from known distributions. In the following, we describe two mechanisms: Laplace and Gaussian mechanisms. These mechanisms add noise $\eta$ to the output of a function $f : \mathbb{X}^k \to \mathbb{R}$ representing a query.

**Lemma A.5** (Laplace mechanism). *Introduced by Dwork et al. (Dwork et al., 2006b), the Laplace mechanism $\mathcal{M}_{Lap}$ for a function $f : \mathbb{X}^k \to \mathbb{R}$ is defined as $\mathcal{M}_{Lap}(x, f, \epsilon) = f(x) + \eta_{Lap}$ where $x \in \mathbb{X}^k$, $\epsilon \in \mathbb{R}$ and $\eta_{Lap} \sim Lap(0, b)$. If $b \geq \Delta f / \epsilon$, then $\mathcal{M}_{Lap}(x, f, \epsilon)$ is $\epsilon$-DP.*

**Lemma A.6** (Gaussian mechanism). *The Gaussian mechanism $\mathcal{M}_{Gauss}$ for a function $f : \mathbb{X}^k \to \mathbb{R}$ is defined as $\mathcal{M}_{Gauss}(x, f, \epsilon, \delta) = f(x) + \eta_{Gauss}$ where $x \in \mathbb{X}$, $\epsilon, \delta \in \mathbb{R}$ and $\eta_{Gauss} \sim Gauss(0, \sigma^2)$. If $\sigma^2 \geq \frac{2 \ln(1.25/\delta) \cdot (\Delta_2 f)^2}{\epsilon^2}$, then $\mathcal{M}_{Gauss}(x, f, \epsilon)$ is $(\epsilon, \delta)$-DP.*

**Local vs central DP**    Let us compare the different DP models:

- The Local DP setting requires the least trust (the aggregator is untrusted). This provides the strongest privacy but suffers from a high error rate of $O(1/n)$.

- Central DP assumes a trusted aggregator who receives raw data of the parties and adds appropriate DP noise, achieving an $O(1/n^2)$ error rate.

- Shuffled DP is an intermediate model that improves upon LDP utility but often relies on the assumption of trusted shufflers to anonymize the inputs when send to the untrusted aggregator.

Our protocol is designed to achieve utility comparable to central DP without requiring a trusted aggregator. By using MPC, we provide a "Distributed-Trust" model where: we match the error rate of central DP and the trust model is distributed across parties. More precisely, we generate a shared random noise $\eta \in \mathbb{R}$, i.e., each party $P_i$ holds a share $[\![\eta]\!]_i$ such that $\text{Rec}(\{[\![\eta]\!]_i\}_{i \in [n]}) = \eta$ and no subset of colluding parties learns any information about $\eta$. The shared noise $\eta$ is drawn either from a Laplace distribution or a Gaussian distribution depending on the mechanism chosen. To generate a shared noise, several papers provide solutions (Eigner et al., 2014; Keller et al., 2024; Sabater et al., 2023). A non-exhaustive description of the existing protocols for drawing shared noise can be found in Appendix B.

### A.5. Examples of U-statistics

We list below a non-exhaustive list of U-statistics of kernel degree 2 and 3:

**Kendall's $\tau$ coefficient**  In the field of statistics, the Kendall rank correlation coefficient (Kendall, 1948) measures the rank correlation of two quantities. Let $\{(x_i, y_i)\}_{i \in [n]}$ be a set of observation. A pair of observation $(x_i, y_i)$ and $(x_j, y_j)$ for $i, j \in [n]$ is said to be *discordant* if the following holds

$$(x_i < x_j \wedge y_i > y_j) \vee (x_i > x_j \wedge y_i < y_j). \tag{7}$$

The Kendall's $\tau$ coefficient uses multiple comparisons as it can be defined as

$$
\begin{aligned}
\tau &= 1 - \frac{2(\text{number of discordant pairs})}{\binom{n}{2}} \\
&= \frac{1}{\binom{n}{2}} \sum_{i<j} \text{sign}(x_i - x_j)\text{sign}(y_i - y_j).
\end{aligned}
\tag{8}
$$

Under the MPC preprocessing paradigm, parties generate pseudorandom correlations before the main computation (see Appendix A.3). By using FSS, the sign function can be evaluated with optimal online communication cost — requiring only a single round and one call of the reconstruction algorithm Rec (see (Boyle et al., 2019) and (Boyle et al., 2016)). However, this efficiency comes at the cost of a heavy offline phase; each FSS key requires $4\ell(\lambda + 1)$ bits where $\lambda \approx 128$ is the security parameter and $\ell = \log_2 |\mathbb{F}|$. Ultimately, computing Kendall's $\tau$ in this framework requires two FSS instantiations and one Beaver Triple.

**Gini Mean Difference**  The Gini Mean Difference (Gini, 1912) is a measure of dispersion that can be expressed as the average of the absolute difference of two variables:

$$
\begin{aligned}
GMD &= \frac{2}{n(n-1)} \sum_{1 \le i < j \le n} |x_i - x_j| \\
&= \frac{2}{n(n-1)} \sum_{1 \le i < j \le n} (2 \cdot \mathbb{I}[x_i > x_j] - 1)(x_i - x_j)
\end{aligned}
\tag{9}
$$

When using the preprocessing paradigm, it requires two FSS instantiations and a single Beaver Triple to compute the kernel function.

**Area Under the ROC Curve (AUC)**  In machine learning and more precisely in binary classification, the AUC-ROC curve is often used to evaluate the performance of a model. It specifies the area under the Receiver Operating Characteristic curve (ROC) which is computed using the True Positive Rate (TPR) and the False Positive Rate (FPR) at various threshold settings of the classifier. The AUC can be interpreted as the probability that the classifier ranks a randomly chosen positive instance higher than a randomly chosen negative instance. Let $\{z_i = (x_i, y_i)\}_{i=1}^n$ be a dataset where $x_i \in \mathbb{X}$ is a data point and $y_i \in \{-1, 1\}$ is its label. Let $n_+$ (resp. $n_-$) be the number of positive (resp. negative) instances, i.e., $n_+ = |\{i \in [n] : y_i = 1\}|$ (resp. $n_- = |\{i \in [n] | y_i = -1\}|$). Let $g$ be the score function that assigns $x_i$ to the confidence probability that $x_i$ belongs to its predicted class and let $x_i^+$ (resp. $x_i^-$) be the $i-th$ instance that is a positive (resp. negative) instance for $i \in [n_+]$ (resp. $i \in [n_-]$). Then, the AUC is expressed as:

$$AUC = \frac{1}{n_- n_+} \sum_{i=1}^{n_+} \sum_{j=1}^{n_-} \mathbb{I}[g(x_i^+) > g(x_j^-)] \tag{10}$$

which can be transformed to the U-statistic form with kernel function $f(z_i, z_j) = \mathbb{I}[x_i > x_j \wedge g(x_i) > g(x_j)] + \mathbb{I}[x_i < x_j \wedge g(x_i) < g(x_j)]$ which can be computed using four comparisons and two multiplications.

**Rand Index**  The Rand Index is a measure of similarity of two data clusterings. Let $D = \{x_i\}_{i \in [n]} \in \mathbb{X}$ be the set of data points. Let $C_1, C_2 : \mathbb{X} \to [m]$ be two classification algorithms where $m$ is the number of classes. The Rand Index $R$ is defined as:

$$R = \frac{1}{\binom{n}{2}} \sum_{1 \le i < j \le n} \mathbb{I}\left[\mathbb{I}[C_1(x_i) = C_1(x_j)] = \mathbb{I}[C_2(x_i) = C_2(x_j)]\right] \tag{11}$$

Under the preprocessing paradigm, 3 FSS instantiations is required to compute a single evaluation of the kernel function.

**Duplicate Pair Ratio**   The Duplicate Pair Ratio is a U-statistic of kernel degree $k = 2$ that measures the proportion of paired observations that are identical across a dataset. Let $D = \{x_i\}_{i \in [n]} \in \mathbb{X}$ be a set of categorical data points. The statistic is defined as:

$$DPR = \frac{1}{\binom{n}{2}} \sum_{1 \leq i < j \leq n} \mathbb{I}\left[x_i = x_j\right]. \tag{12}$$

Computing the kernel function of this statistic requires only a single invocation of FSS.

**Triangle Counting**   The Triangle Counting is a problem introduced in the graph theory litterature and yields a U-statistic of degree 3. Given a graph $G = (W, F)$, the goal is to compute the number of triangles, i.e., the number $T_G$ of unordered triples $(x_1, x_2, x_3)$ such that $(x_1, x_2), (x_2, x_3), (x_1, x_3) \in F$. The number $T_G$ can be expressed as a U-statistic with kernel of degree $k = 3$, i.e.,

$$T_G = \sum_{(x_1, x_2, x_3) \in C_3^{|W|}} \mathbb{I}[(x_1, x_2) \in F] \cdot \mathbb{I}[(x_2, x_3) \in F] \cdot \mathbb{I}[(x_1, x_3) \in F].$$

This requires every data owner $w \in W$ to store the list of vertices $w' \in W$ which are adjacent to $w$, i.e., $(w, w') \in F$. Computing such a kernel function requires 3 FSS invocations in the secret-shared domain.

# B. Drawing distributed noise

We discuss some strategies to generate secret shared noise from a Laplace or Gaussian distribution. For the simplicity of our explanation, we make abstraction of some discretization details, see (Canonne et al., 2020; Eigner et al., 2014) for a more in-depth discussion of elements to take into account when using discretized distributions for differential privacy.

## B.1. Laplace Distribution

### B.1.1. EXISTING APPROACHES

In (Sabater et al., 2023), to draw from a Laplace distribution of parameters $(0, b)$, the authors rely on the inverse cumulative distribution function of the Laplace distribution which can be expressed as:

$$F^{-1}(p) = -b \cdot \text{sign}(p - 0.5) \ln(1 - |p - 0.5|) \tag{13}$$

where $| \cdot |$ is the absolute value. To compute the $\ln$, they use the Cordic algorithm which is an iterative algorithm that requires only additions, bit shifts and comparisons. The distributed version of the Cordic algorithm needs $O(\ell^2 \log \ell)$ secure multiplications in $O(\ell^2)$ rounds if we assume that the number of iterations to converge is of order $O(\ell)$. The protocol requires to draw randomly $a'$ uniformly from $[0, L]$ such that $a = -b \cdot \text{sign}(1 - a'/L)$. Then, we can multiply $a$ by a random sign $s$ drawn uniformy from $\{-1, 1\}$. Overall, the communication complexity required to draw from $\text{Lap}(0, b)$ using (Sabater et al., 2023) is $O(\ell^2 \log \ell)$ secure multiplications in $O(\ell^2)$ rounds if $L = 2^\ell$.

In (Eigner et al., 2014), the authors distributively sample from $\text{Lap}(0, \frac{\Delta f}{\epsilon})$ by noting that $\text{Lap}(0, m) = \text{Exp}(\frac{1}{m}) - \text{Exp}(\frac{1}{m})$ such that $\text{Exp}(m') = \frac{-\ln x}{m'}$ where $x \sim \text{Uni}((0, 1])$ which reduces the complexity of the protocol to sample uniformly and then computing $\ln$ in a distributed manner. This last operation is quite costly as it requires $O(\ell^2)$ secure multiplications in $O(\ell)$ rounds.

Balle et al. (Balle et al., 2020) propose a protocol for privately computing a sum in shuffled model. Each party $i \in [n]$ holds an input $x_i \in [0, 1]$. To generate the required DP noise in a distributed way, the parties jointly sample from a discrete Laplace distribution: they use the fact that the sum of $n$ independent differences of two Polya random variables is exactly a discrete Laplace random variable, i.e., if $X_i$ and $Y_i$ are independent $\text{Polya}(1/n, e^{-1/b})$ random variables where $b = \Delta f/\epsilon$, then, $Z = \sum_{i \in [n]} X_i - Y_i$ follows $\text{DLap}(e^{-1/b})$. After adding their local noise shares, the parties send secret shares of their noisy contributions to the shufflers. The aggregator then reconstructs the total value and obtains a differentially private estimate of the sum via the discrete Laplace mechanism.

### B.1.2. ADAPTATION OF BALLE ET AL. (BALLE ET AL., 2020)

We observe that the protocol from Balle et al. (Balle et al., 2020) can realize the functionality $\mathcal{F}_{\text{noise}}$ for $(\epsilon, 0)$-DP in the semi-honest model. However, because our protocol $\prod_{\text{U-MPC}}$ operates over the finite field $\mathbb{F}_{2^\ell}$, the noise must remain below

$2^\ell$ to avoid modular wrap-around.

Assuming that $p >> n$, Protocol B.1 generates distributed discrete Laplace noise from Balle et al. (Balle et al., 2020). This protocol requires $n^2 \cdot \ell$ bits of communication.

---

**Protocol $\prod_{\text{DLap}}$**

**Input:**

- $\epsilon > 0$: DP parameter
- $s$: sensitivity parameter

**Online phase:**

1. Party $P_i$ draws $a_i, b_i \sim \text{Polya}(1/n, e^{-\epsilon/s})$, for $i \in [n]$.

2. Party $P_i$ creates shares $[\![a_i]\!] \leftarrow \text{Share}(n, n, a_i)$ and sends to each party $P_j$ the share $[\![a_i]\!]_j$, for $i, j \in [n]$.

3. Same for $b_i$, for $i \in [n]$.

4. Party $P_i$ computes $c_i = \sum_{j \in [n]} [\![a_j]\!]_i - [\![b_j]\!]_i$.

5. Party $P_i$ returns $c_i$ for $i \in [n]$.

---

*Protocol B.1.* Protocol $\prod_{\text{DLap}}$ to generate distributed discrete Laplace noise from (Balle et al., 2020).

*Remark* B.1 (Local distributed noise generation). Protocol B.1 can be modified to use only local computations (discard Steps (2) and (3)) if we assume that each party adds its DP noise share to values that are already uniformly distributed. In Protocol $\prod_{\text{U-MPC}}$, during the *Noise addition phase*, each party adds its share of the distributed noise $\eta$ to the local sum of shares. Because the convolution of a uniform distribution with any independent distribution is itself uniform, the resulting $z_i$ remains uniform distributed over $\mathbb{F}_{2^\ell}$ assuming no wrap around.

## B.2. Gaussian Distribution

### B.2.1. EXISTING APPROACHES

In (Sabater et al., 2023), the authors consider several techniques to compute a shared noise from the Gaussian distribution. We describe in the following two methods:

- the Central Limit Theorem (CLT) approach introduced by (Dwork et al., 2006a). To generate shares of $x \sim \text{Gauss}(0, n/4)$ where $n$ is both the number of parties and the number of coin flips. Each party $P_i$ randomly selects $c_i \leftarrow \{0, 1\}$ and shares it to the other parties. Party $P_i$ gets shares $[\![c_1]\!]_i, \ldots, [\![c_n]\!]_i$. Since the parties can deviate from the intended protocol (due to the malicious setting), the bits $c_i$ are considered low quality bits as these bits could have been chosen adversarially. Then, imagine having a public source of random bits $[\![b_1]\!], \ldots, [\![b_n]\!]$, parties can convert the low quality bits $c_i$ into high quality bits by applying $c_i \oplus b_i$ for $i \in [n]$. Finally, each party can sum their shares such that $[\![x]\!]_i = \sum_{j=1}^n 2([\![c_j]\!]_i \oplus [\![b_j]\!]_i) - 1$ to get $x \in \{-1, 1\}$. This is similar to drawing $n$ times an unbiased coin.

  Communication complexity-wise, computing shares of elements from $\{0, 1\}$ requires two secure multiplications. If $\ell = \log_2 |\mathbb{F}|$ when working in $\mathbb{F}$, the communication complexity for one party would require to send $O(n\ell)$ bits in constant rounds without counting the source of public randomness.

- The polar version of Box-Muller generates a pair of random numbers that follows a Gaussian distribution using a uniform source of randomness. Let $(x, y)$ be a pair of random samples uniformly drawn from $(-1, 1)$. Let $U = x^2 + y^2$. We draw pairs $(x, y)$ until $0 < U \leq 1$. Then, $z_1 = x\sqrt{\frac{-2 \ln U}{U}}$ and $z_2 = y\sqrt{\frac{-2 \ln U}{U}}$ are sampled from $\text{Gauss}(0, 1)$. To extend this method to the secret sharing setting, the main bottleneck resides in the fact that parties need to draw distributively and uniformly over $(-1, 1)$ in the share domain over $\mathbb{F}$. Such an approach has been suggested by (Sabater et al., 2023).

# C. Analysis of Protocol 3.1

We now provide details on the properties described in Sec 4.

## C.1. Correctness

**Lemma C.1** (Correctness). *The protocol* $\prod_{U\text{-}MPC}$ *computes the incomplete U-statistic* $\hat{U}_{f,E}$ *with additional noise calibrated to the privacy parameters* $(\epsilon, \delta)$.

*Proof.* When reconstructing, the aggregator computes:

$$
\begin{aligned}
\prod_{\text{U-MPC}} \left([n], E, \epsilon, \delta, \Delta f, \{x_i\}_{i \in [n]}\right) &= \frac{1}{|E|} \mathsf{Rec}(\{z_i\}_{i \in [n]}) \\
&= \frac{1}{|E|} \Big( \eta + \sum_{e \in E} \sum_{j \in [k]} [\![f(x_e)]\!]_{e_j} \Big) \\
&= \frac{1}{|E|} \Big( \eta + \sum_{e \in E} f(x_e) \Big) \\
&= \frac{\eta}{|E|} + U_{f,E}
\end{aligned}
$$

$\square$

In this equation, the step $\sum_{j \in [k]} [\![f(x_e)]\!]_{e_j} = f(x_e)$ follows from the properties of the additive secret sharing scheme, which reconstructs a secret by simply adding the shares.

## C.2. Privacy and security

Next, we outline the privacy guarantees and why our scheme remains secure in the presence of a semi-honest static adversary (see the threat model in Sec 3).

### C.2.1. INFORMAL SECURITY

Let us start with the computation of $f(e)$ for a single edge $e \in E$. The $k$ involved parties use additive secret sharing, which is a $k$-threshold secret sharing scheme and hence is secure under a honest majority setting. There are basically two cases: either all $k$ parties in $e$ are corrupted or not all $k$ parties in $e$ are corrupted. If all parties in $e$ are corrupted, they can collude and reveal $f(e)$, however since they are corrupted parties and collude they could determine $f(e)$ even without being running our protocol. If not all parties in $e$ are corrupted, they can't reconstruct $f(e)$.

Next, phase 3, the noise generating phase, mainly uses a functionality $\mathcal{F}_{\text{noise}}$, for which we required that a secure implementations is provided.

The aggregation phase is secure as it is an addition using additive secret sharing. Every party owns a number $z_i$, and any set of $n-1$ of these numbers are uniformly randomly distributed because any set of $n-1$ shares $[\![\eta]\!]_i$ is uniformly randomly distributed. Hence, from the numbers $z_i$ nothing can be inferred by the adversary.

### C.2.2. FORMAL SECURITY PROOF

We prove our result using the universal composability (UC) paradigm (Canetti, 2001). Within the UC framework, security is defined by comparing two executions. In the ideal-world execution of a function $g$, parties transmit their inputs $x_1, \ldots, x_n$ to a trusted third party, which evaluates the function $g$ and returns the outputs $g(x_1, \ldots, x_n)$ to the parties. Demonstrating the security of a protocol then amounts to proving that the real-world execution (where we execute our protocol) is indistinguishable from this ideal-world execution. In the hybrid model of the UC framework, the real-world execution have access to specific ideal functionalities, in this case $\mathcal{F}_f$ and $\mathcal{F}_{\text{noise}}$.

**Lemma C.2** (Privacy and security guarantees). *Let* $\mathcal{A}$ *be a static semi-honest adversary which corrupts at most* $n-1$ *parties (the dishonest majority model* $\mathcal{M}_{Dis}$*). Let* $\mathcal{F}_f$ *be the functionality to compute* $f$ *and* $\mathcal{F}_{noise}$ *the functionality to compute the shared noise. Our protocol* $\prod_{U\text{-}MPC}$ *is secure against* $\mathcal{A}$ *in the* $(\mathcal{F}_f, \mathcal{F}_{noise})$*-hybrid model and additionally satisfies central* $(\epsilon, \delta)$*-differential privacy.*

*Proof.* We aim to show that there exists a simulator $\mathcal{S}$, given access to the ideal functionalities $\mathcal{F}_f$ and $\mathcal{F}_{\text{noise}}$, that can generate an indistinguishable transcript from the one produced during the real-world execution of the protocol.

Let $\mathcal{S}$ be a simulator. Let $H \subseteq [n]$ be the set of honest parties such that $1 \leq |H| \leq n$. $\mathcal{S}$ receives the computed U-statistic $\hat{U}_{f,E}$ from the ideal functionality. During the *Sharing phase*, parties distribute their respective inputs using $(k, k)$-threshold SS. Let $e = (e_1, \ldots, e_k) \in E$ be an edge of the graph $G$. There are two cases to consider:

1. all parties on edge $e$ are corrupted, i.e., $|H \cap e| = 0$. In that case, the corrupted parties $P_{e_1}, \ldots, P_{e_k}$ can communicate following the communication pattern imposed by the adversary $\mathcal{A}$ and the simulator $\mathcal{S}$ does not intervene. Then, during the *Computing phase*, $\mathcal{S}$ invokes $\mathcal{F}_f$ on the inputs of the corrupted parties and returns the output to the adversary. For $j \in [k]$, each party $P_j$ on $e$ computes $y_j$ accordingly.

2. Edge $e$ includes $d_e = |H \cap e| \geq 1$ honest parties. Since the simulator $\mathcal{S}$ does not know the honest parties' inputs $x_j$, it instead generates shares of a random value $r_e \in \mathbb{F}$ and distributes them to the $k - d_e$ other parties. By definition of $(k, k)$-threshold SS, any $k - 1$ shares is uniformly distributed over $\mathbb{F}$. During the *Computing phase*, the simulator $\mathcal{S}$ selects another random value $r'_e \in \mathbb{F}$, secret-shares it among the parties on edge $e$. Since the adversary observes $k - d_e < k$ shares, it cannot distinguish between the simulated shares from valid shares of the real computation result $f(x_e)$. Then, each party $P_i$ computes $y_i = \sum_{e \in E_{\{i\}}} [\![r'_e]\!]_i$.

During the *Noise addition phase*, $\mathcal{S}$ simulates the functionality $\mathcal{F}_{\text{noise}}$ by sending to each party $P_i$ the share $\eta_i = [\![(\hat{U}_{f,E} - \sum_{e \in E} r'_e)/|E|]\!]_i$ for $i \in [n]$. Each party $P_i$ computes $z_i = y_i + |E| \cdot \eta_i$ and send it to the aggregator. Hence, the aggregator computes

$$
\begin{aligned}
z &= \mathsf{Rec}(\{y_i\}_{i \in [n]}) + \mathsf{Rec}(\{|E| \cdot \eta_i\}_{i \in [n]}) \\
&= \sum_{e \in E} r'_e + |E| \cdot (\hat{U}_{f,E} - \sum_{e \in E} r'_e)/|E| \\
&= \hat{U}_{f,E}.
\end{aligned}
$$

At the end, the view of the corrupted parties in the real world is indistinguishable from the simulation that $\mathcal{S}$ produces.

In terms of privacy, the algorithm satisfies central DP due to the noise $\eta$ introduced during the *Noise addition phase*, details are provided in Sec C.2.3.

$\square$

### C.2.3. PRIVACY

There remains to show that $\eta/|E|$ is the appropriate amount of noise. To establish this, let us consider the sensitivity $\Delta U_{f,E}$. Applying the definition of sensitivity,

$$
\Delta U_{f,E} = \max\{\|U_{f,E}(x) - U_{f,E}(x')\| \mid x, x' \in \mathbb{X}^* \wedge \mathrm{adjacent}(x, x')\}
$$

Now consider adjacent datasets $x$ and $x'$. Let $i \in [n]$ such that the $i$-th instance is the only one in which $x$ and $x'$ differ, i.e., $x_i \neq x'_i$ and $\forall j \neq i : x_j = x'_j$. Then, for an edge $e \in E$, if $i \in e$ there holds $\|f(x_e) - f(x'_e)\| \leq \Delta f$ while $i \notin e$ there holds $\|f(x_e) - f(x'_e)\| = 0$. There follows

$$
\begin{aligned}
\|U_{f,E}(x) - U_{f,E}(x')\| &\leq \frac{1}{|E|} \sum_{e \in E} \|f(x_e) - f(x'_e)\| \\
&= \frac{1}{|E|} \sum_{e \in E_{\{i\}}} \|f(x_e) - f(x'_e)\| \\
&\leq \frac{1}{|E|} \delta_G^{max} \Delta f \\
&= \frac{\delta_G^{max} \Delta f}{|E|}
\end{aligned}
$$

Hence, $\Delta U_{f,E} = \delta_G^{max} \Delta f / |E|$. As the protocol ensures that a noise term $\eta$ is sufficient to privatize a function with sensitivity $\delta_G^{max} \Delta f$, a noise term $\eta/|E|$ is sufficient to privatize a function with sensitivity $\delta_G^{max} \Delta f / |E|$ such as $U_{f,E}$.

## C.3. Communication complexity

We recall that we compute $\hat{U}_{f,E}$ in the preprocessing paradigm (see subsection A.3).

**Lemma C.3** (Communication complexity). *Let* $\mathsf{C}_\eta^C$, $\mathsf{C}_\eta^R$ *be respectively the number of bits exchanged and the number of rounds to generate a shared noise. Let* $\mathsf{C}_f^C$, $\mathsf{C}_f^R$ *be respectively the number of bits exchanged and the number of rounds during the online phase for the computation of* $f$. *In the online phase, the protocol* $\prod_{U\text{-}MPC}$ *requires* $|E|k(k-1)\ell + |E| \cdot \mathsf{C}_f^C + \mathsf{C}_\eta^C + n\ell = O(|E|(k^2\ell + \mathsf{C}_f^C) + \mathsf{C}_\eta^C)$ *bits of communication in* $\max(\mathsf{C}_f^R, \mathsf{C}_\eta^R) + 2$ *rounds.*

*Proof.* We measure communication complexity in terms of bits exchanged phase by phase.

1. In the *Sharing phase*, for every of the $|E|$ edges, each of the $k$ involved parties has to send $k-1$ shares of size $\ell$ bits to the other parties in the same edge. Hence, this step requires $|E|k(k-1)\ell$ bits. The sharing can be done in parallel, hence it requires only one round.

2. The communication complexity of the *Computing phase* depends on the kernel function $f$. It also depends on the number of edges in $E$. Hence, it requires $|E| \cdot \mathsf{C}_f^C$ bits. Since each evaluation of $f$ can be parallelized, it requires $\mathsf{C}_f^R$ rounds of communication.

3. The *Noise addition phase* creates a shared noise according to some chosen protocol. It requires an exchange of $\mathsf{C}_\eta^C$ bits in $\mathsf{C}_\eta^R$ rounds.

4. The last step requires each of the $n$ parties to send $\ell$ bits to the aggregator in one round.

$\square$

Since constructing a communication-efficient offline is of independent interest, we only give the communication complexity of the online phase. Please refer to (Abram & Scholl, 2022; Frederiksen et al., 2015; Boyle et al., 2020) for efficient protocols to generate correlated randomness.

# D. Comparative Evaluation

## D.1. Details and Proofs for MSE

Let the population quantity be
$$U_f = \mathbb{E}_{X_1, X_2 \sim P_{\mathbb{X}}}[f(X_1, X_2)].$$

The U-statistic $U_{f,C_k^n}$ defined in Definition 2.1 is an unbiased estimator of $U_f$. While Bell et al. (Bell et al., 2020) analyze the error between their estimator and the population quantity $U_f$, our focus is instead on the error between the estimator and the sample U-statistic $U_{f,C_2^n}$.

We provide in the following a detailed focus on the MSE of each protocol under $\epsilon$-DP for pairwise data, i.e., $k = 2$ assuming that $\mathbb{Y} = [0,1]$ and $f$ is $L_f$-Lipschitz.

### D.1.1. BELL

To estimate the population statistic $U_f$, Bell et al. (Bell et al., 2020) compute a DP unbiased estimator of $U_f$ using a sample of size $n$. First, the population statistic $U_f$ is approximated by evaluating it on a finite sample, in particular

$$U_{f,C_2^n} = \binom{n}{2}^{-1} \sum_{(x_1,x_2) \in C_2^n} f(x_1, x_2)$$

is a sum over all $\binom{n}{2}$ pairs $(x_1, x_2)$ of a sample of size $n$, Next, $U_{f,C_2^n}$ is approximated by discretizing the data, in particular

$$U_{f,C_2^n,\pi} = \binom{n}{2}^{-1} \sum_{(x_1,x_2) \in C_2^n} f_A(\pi(x_1), \pi(x_2))$$

where $\pi : \mathbb{X} \to [t]$ is a discretization function and $f_A(i, j) = e_i^\top A e_j$ where $e_i \in \mathbb{R}^t$ is a vector with a 1 on position $i$ and 0 elsewhere, and $A$ is a matrix with $A_{i,j} = f(\gamma(i), \gamma(j))$ where $\gamma(i) \in \pi^{-1}(i)$ is a representative for the $i$-th bin of the discretization $\pi$. Finally,

$$\bar{U}_f = \binom{n}{2}^{-1} \sum_{(x_1, x_2) \in C_2^n} \hat{f}_A(\mathcal{R}(\pi(x_1)), \mathcal{R}(\pi(x_2)))$$

where $\mathcal{R} : [t] \to [t]$ is an $\epsilon$-LDP randomizer and $\hat{f}_A$ is a function such that $\hat{f}_A(\mathcal{R}(\pi(x_1)), \mathcal{R}(\pi(x_2)))$ is an unbiased estimator of $f_A(\pi(x_1), \pi(x_2))$. In particular, $\mathcal{R} : [t] \to [t]$ is a randomizer which returns a uniformly distributed element from $[t]$ with probability $\beta$ and returns its input with probability $(1 - \beta)$, i.e., for all $x, y \in [t]$

$$\mathbb{P}\left[\mathcal{R}(x) = y\right] = \beta/t + (1 - \beta)\mathbb{I}[x = y],$$

and

$$\hat{f}_A(i, j) = (1 - \beta)^{-2}(e_i - b)^\top A(e_j - b)$$

where $b \in \mathbb{R}^t$ is a vector with in every component $\beta/t$.

Comparing to the population statistic $U_f$ the mean squared error $E_{tot} = \mathbb{E}\left[(U_f - \bar{U}_f)^2\right]$ can be decomposed as:

$$E_{tot} = E_{sample} + E_{discr} + E_{DP}$$

with

$$
\begin{aligned}
E_{sample} &= \mathbb{E}_{x_1 \ldots x_n}\left[(U_f - U_{f, C_2^n})^2\right] \\
E_{discr} &= (U_{f, C_2^n} - U_{f, C_2^n, \pi})^2 \\
E_{DP} &= \mathbb{E}_\mathcal{R}\left[(U_{f, C_2^n, \pi} - \bar{U}_f)^2\right]
\end{aligned}
$$

Let us now derive or review bounds on each of these errors.

**The effect of local DP**  Bell et al. (Bell et al., 2020, Appendix A.1) prove that if $\forall x, y : f(x, y) \in [0, 1]$:

$$E_{DP} \leq \left(\frac{1}{n(1 - \beta)^2} + \frac{(1 + \beta)^2}{2n(n - 1)(1 - \beta)^4}\right) \tag{14}$$

To achieve $\epsilon$-LDP, we need that $\mathbb{P}\left[\mathcal{R}(i) = i\right] \leq e^\epsilon \mathbb{P}(\mathcal{R}(i) = j)$. We have that $\mathbb{P}(\mathcal{R}(i) = i) = (1 - \beta) + \beta/t$ and for $j \neq i$ that $\mathbb{P}(\mathcal{R}(i) = j) = \beta/t$, hence there must hold $(1 - \beta) + \beta/t \leq e^\epsilon(\beta/t)$ which is equivalent to $\beta \geq ((e^\epsilon - 1)/t + 1)^{-1}$. This condition is satisfied if there holds $\beta \geq ((\epsilon/t + 1)^{-1})$. Combining this with Eq (14) one can see that for large $t$ and small $\epsilon$ we have approximately

$$E_{DP} \approx \frac{t^2}{n\epsilon^2} \tag{15}$$

**The effect of discretization**  We have that

$$U_{f, C_2^n, \pi} = \frac{1}{\binom{n}{2}} \sum_{(i,j) \in C_2^n} f(\gamma(\pi(x_i)), \gamma(\pi(x_j)))$$

and

$$U_{f, C_2^n} = \frac{1}{\binom{n}{2}} \sum_{(i,j) \in C_2^n} f(x_i, x_j)$$

Let $f$ be $L_f$-Lipschitz, i.e., $\forall x, x', y, y' : |f(x, y) - f(x', y')| \leq L_f(|x - x'| + |y - y'|)$. Moreover, let us assume that $\forall x : f(x, y) \in [0, 1]$, $\pi(x) = \lceil xt + 1/2 \rceil$ and $\gamma(i) = i - \frac{1}{2}$. Then, for all $i, j \in [n]$:

$$|f(x_i, x_j) - f(\gamma(\pi(x_i)), \gamma(\pi(x_j)))| \leq \frac{L_f}{t}$$

There follows, $\left|U_{f, C_2^n, \pi} - U_{f, C_2^n}\right| \leq \frac{L_f}{t}$, and

$$E_{disc} \leq \frac{L_f^2}{t^2} \tag{16}$$

**The effect of working with a finite sample**   This can be computed using the variance of Hoeffding (Hoeffding, 1992):

$$\text{Var}(U_f, U_{f,C_2^n}) = \frac{2}{n(n-1)}(2(n-2)\sigma_1 + \sigma_2)$$

where $\sigma_1 = \underset{x_1}{\text{Var}}(\underset{x_2}{\mathbb{E}}(f(x_1, x_2)))$ and $\sigma_2 = \underset{x_1,x_2}{\text{Var}}(f(x_1, x_2))$.

At the end, the error between the estimator and the U-statistic sample is defined as:

$$E_{tot} - E_{sample} = E_{DP} + E_{discr}$$
$$\leq \frac{t^2}{n\epsilon^2} + \frac{L_f^2}{t^2}. \tag{17}$$

### D.1.2. OUR PROTOCOL

Our Protocol 3.1 computes an unbiased estimator of the population statistic $U_f$.

1. First, the population statistic $U_f$ is approximated by using a sample $\{x_i\}_{i\in[n]}$ to obtain $U_{f,C_2^n}$.

2. Then, $U_{f,C_2^n}$ is approximated by sampling a set $E \subset C_2^n$. In particular,

$$U_{f,E} = \frac{1}{|E|} \sum_{(i,j)\in E} f(x_i, x_j)$$

   is an unbiased estimator of $U_{f,C_2^n}$.

3. At the end, the protocol 3.1 uses the Laplace mechanism (see Lemma A.5) to obtain

$$\hat{U}_{f,E} = U_{f,E} + \eta$$

   where $\eta \sim Lap(0, s/\epsilon)$ with $s = \delta_G^{max}\Delta f$ determined in Appendix C.

We write the total MSE for our protocol as

$$E_{tot} = E_{sample} + E_{inc} + E_{DP}$$

where

$$
\begin{aligned}
E_{sample} &= \mathbb{E}_{x_1\ldots x_n}\left[(U_f - U_{f,C_2^n})^2\right] \\
E_{inc} &= \mathbb{E}_E\left[(U_{f,C_2^n} - U_{f,E})^2\right] \\
E_{DP} &= \mathbb{E}_{\mathcal{R}}\left[(U_{f,E} - \hat{U}_{f,E})^2\right]
\end{aligned}
$$

Let us bound each of these error terms.

**The effect of working with an incomplete U-statistic**   Let us compare the incomplete U-statistic $U_{f,E}$ to the U-statistic of the sample $U_{f,C_2^n}$. While it is more difficult to accurately write a closed form expression for $E_{inc}$ if $E$ is sampled using BalancedSamp (Algorithm 1) than if it is sampled uniformly, we can see $U_{f,E}$ as the mean of a stratefied sample, implying that its variance compared to $U_{f,C_2^n}$ is smaller than the corresponding statistic with a uniformly sampled $E$. We can hence bound $E_{inc}$ by the MSE $E'_{inc}$ induced when $E$ is sampled uniformly over $C_2^n$.

*Claim* D.1.  Under the sampling procedure BalancedSamp (Algorithm 1) for the set $E$ and $\mathbb{Y} \in [0, 1]$, let $N = \begin{pmatrix} n \\ 2 \end{pmatrix}$ and $m = |E|$ such that

$$E_{inc} \leq \frac{N - m}{4m(N - 1)}.$$

*Proof.* We begin by considering $E'_{inc} = \mathbb{E}[(U_{f,C_2^n} - U_{f,E})^2]$ where $E$ is sampled without replacement, i.e., $E$ is drawn uniformly at random from all subsets of $C_2^n$ of size $m$. This provides an upper bound since $E_{inc} \leq E'_{inc}$.

Let $I_v$ be the indicator that the pair $v$ is included in $E$, i.e., $I_v = 1$ if $v \in E$, else 0. We have, for $v \in C_2^n$, $\mathbb{P}[I_v = 1] = \frac{m}{N}$ so that $\mathrm{Var}(I_v) = \frac{m}{N} - \frac{m^2}{N^2} = \frac{m(N-m)}{N^2}$ and for $v, v' \in C_2^n$ with $v \neq v$, $\mathbb{P}[I_v = 1, I_{v'} = 1] = \frac{m(m-1)}{N(N-1)}$. Then, we can deduce that for $v, v' \in C_2^n$, $v \neq v$,

$$
\begin{aligned}
\mathrm{Cov}(I_v, I_{v'}) &= \mathbb{E}[I_v I_{v'}] - \mathbb{E}[I_v]\mathbb{E}[I_{v'}] \\
&= \frac{m(m-1)}{N(N-1)} - \frac{m^2}{N^2} \\
&= \frac{m(m-N)}{N^2(N-1)}.
\end{aligned}
$$

Let us first prove the following statement. Let $S_1 = \sum_{v \in C_2^n} f(x_v)$ and $S_2 = \sum_{v \in C_2^n} f(x_v)^2$. Let $\mu = \frac{1}{N} \sum_{v \in C_2^n} f(x_v)$. Then, we have that: $S_2 - \frac{1}{N}S_1^2 = \sum_{v \in C_2^n} (f(x_v) - \mu)^2$.

To prove this statement, let us develop the term $\sum_{v \in C_2^n} (f(x_v) - \mu)^2$.

$$
\begin{aligned}
\sum_{v \in C_2^n} (f(x_v) - \mu)^2 &= \sum_{v \in C_2^n} f(x_v)^2 - 2f(x_v)\mu + N\mu^2 \\
&= S_2 - \frac{2}{N}S_1^2 + N\frac{S_1^2}{N^2} \\
&= S_2 - \frac{1}{N}S_1^2.
\end{aligned}
\tag{18}
$$

Finally, we can compute the variance as follows:

$$
\begin{aligned}
&\mathrm{Var}_{E|x_1 \ldots x_n}(U_{f,E}) \\
&= \mathrm{Var}_{E|x_1 \ldots x_n}\left(\frac{1}{m} \sum_{v \in E} f(x_v)\right) \\
&= \mathrm{Var}_{E|x_1 \ldots x_n}\left(\frac{1}{m} \sum_{v \in C_2^n} I_v f(x_v)\right) \\
&= \frac{1}{m^2}\left(\sum_{v \in C_2^n} f(x_v)^2 \mathrm{Var}(I_v) + \sum_{\substack{v,v' \in C_2^n \\ v \neq v'}} \mathrm{Cov}(I_v, I_{v'})f(x_v)f(x_{v'})\right) \\
&= \frac{1}{m^2}\left(\sum_{v \in C_2^n} \frac{m(N-m)}{N^2} f(x_v)^2 + \sum_{\substack{v,v' \in C_2^n \\ v \neq v'}} \frac{m(m-N)}{N^2(N-1)} f(x_v)f(x_{v'})\right) \\
&= \frac{N-m}{mN^2} \sum_{v \in C_2^n} f(x_v)^2 + \frac{m-N}{mN^2(N-1)} \sum_{\substack{v,v' \in C_2^n \\ v \neq v'}} f(x_v)f(x_{v'}) \\
&= \frac{N-m}{mN^2}\left(S_2 - \frac{S_1^2 - S_2}{N-1}\right) \\
&= \frac{N-m}{mN^2} \cdot \frac{NS_2 - S_1^2}{N-1} \\
&= \frac{N-m}{mN^2} \cdot \frac{N}{N-1} \sum_{v \in C_2^n} (f(x_v) - \mu)^2 \\
&= \frac{N-m}{(N-1)m} \cdot \mathrm{Var}_{X \sim \mathrm{Uni}(C_2^n)}(f(X)).
\end{aligned}
$$

Under the assumption that $\mathbb{Y} \in [0, 1]$, we get that

$$E_{inc} \leq E'_{inc} \leq \frac{N - m}{4m(N - 1)}.$$

□

**The effect of privacy noise** Let $\eta$ be the noise term computed for the Laplace mechanism $\mathcal{M}_{Lap}$, i.e., $\eta \sim Lap(0, b = s/\epsilon)$ where $s = \delta_G^{max} \Delta f$.

*Claim* D.2. When using BalancedSamp (Algorithm 1) for sampling the set $E$ and under central $\epsilon$-DP,

$$E_{DP} \approx 2\frac{(2(\Delta f))^2}{n^2 \epsilon^2}$$
$$= \Theta\left(\frac{1}{n^2 \epsilon^2}\right).$$

*Proof.* Recall that $\eta \sim Lap(0, \delta_G^{max} \Delta f/\epsilon)$. Under the BalancedSamp sampling procedure (Algorithm 1), all vertices have degree $\lceil k|E|/n \rceil$ or $\lfloor k|E|/n \rfloor$. Hence,

$$E_{DP} = \text{Var}\left(\frac{\eta}{|E|}\right)$$
$$= 2\left(\frac{\delta_G^{max} \Delta f}{\epsilon |E|}\right)^2$$
$$\approx \frac{8}{n^2 \epsilon^2}$$

□

**The effect of working with a finite sample** Let $\sigma_1 = \underset{x_1 \sim P_{\mathbb{X}}}{\text{Var}} (\underset{X_2 \sim P_{\mathbb{X}}}{\mathbb{E}} [f(x_1, X_2)])$ and $\sigma_2 = \underset{x_1, x_2 \sim P_{\mathbb{X}}}{\text{Var}} (f(x_1, x_2))$. We can use the result from Hoeffding (Hoeffding, 1992).

$$E_{sample} = \frac{2}{n(n-1)}(2(n-2)\sigma_1 + \sigma_2).$$

Since $\mathbb{Y} = [0, 1]$, we can bound $\sigma_1$ and $\sigma_2$ by $1/4$ and get $E_{sample} \leq \frac{2(2n-3)}{n(n-1)} < 4/n$.

The error between our estimator and the U-statistic sample is expressed as:

$$E_{tot} - E_{sample} = E_{inc} + E_{DP}$$

which gives the following lemma.

**Lemma D.3.** *Assume $f(x, y) \in [0, 1]$ for all $x, y \in \mathbb{X}$. Let $N = \binom{n}{2}$ and let $\Delta f$ be the sensitivity of $f$. Using* BalancedSamp *(Algorithm 1) for sampling the set $E$, the protocol $\prod_{U\text{-MPC}}$ computes the U-statistic $\hat{U}_{f,E}$ under $\epsilon$-DP such that*

$$\text{Var}(\hat{U}_{f,E}) \leq \frac{N - |E|}{4|E|(N - 1)} + 2\frac{(2\Delta f)^2}{n^2 \epsilon^2}.$$

*Proof.* Follows from Claims D.1 and D.2. □

The proof extends to general $k$ by replacing pairs $(i, j)$ with $k$-tuples and applying the same arguments.

### D.1.3. GHAZI

Similar to Bell, Ghazi et al.'s paper computes an unbiased U-statistic by discretizing the data, in particular they consider

$$U_{f,C_2^n,\pi} = \binom{n}{2}^{-1} \sum_{(x_1,x_2)\in C_2^n} f_A(\pi(x_1),\pi(x_2))$$

where $\pi$ and $A$ are defined in Bell's section. The authors approximate this value by using the JL theorem. In particular, let $A = L^T R$ where $L, R \in \mathbb{R}^d$ with $d < t$ such that

$$\tilde{U}_{f,C_2^n,\pi} = \left\langle \frac{1}{n} \sum_{i\in[n]} L e_{\pi(x_i)}, \frac{1}{n} \sum_{i\in[n]} R e_{\pi(x_i)} \right\rangle$$

where $\langle \cdot, \cdot \rangle$ denotes the inner product and $e_i \in \mathbb{R}^t$ denotes the one-hot-vector of size $t$ with a 1 at position $i$. Finally, the authors propose this unbiased estimator:

$$\tilde{U}_f = \left\langle \frac{1}{n} \sum_{i\in[n]} \mathcal{R}(L e_{\pi(x_i)}), \frac{1}{n} \sum_{i\in[n]} \mathcal{R}(R e_{\pi(x_i)}) \right\rangle$$

where $\mathcal{R} : \mathbb{R}^d \to \mathbb{R}^d$ is an $\epsilon$-LDP randomizer from (Błasiok et al., 2019). The MSE can be decomposed into

$$E_{tot} = E_{sample} + E_{discr} + E_{JL} + E_{DP}$$

where

$$
\begin{aligned}
E_{sample} &= \mathbb{E}_{x_1 \ldots x_n} \left[ (U_f - U_{f,C_2^n})^2 \right] \\
E_{discr} &= (U_{f,C_2^n} - U_{f,C_2^n,\pi})^2 \\
E_{JL} &= \mathbb{E}_{A=L^T R} \left[ (U_{f,C_2^n,\pi} - \tilde{U}_{f,C_2^n,\pi})^2 \right] \\
E_{DP} &= \mathbb{E}_{\mathcal{R}} \left[ (\tilde{U}_{f,C_2^n,\pi} - \tilde{U}_f)^2 \right]
\end{aligned}
$$

Let us separately bound these error terms.

**The effect of working with a finite sample**  This is similar to Bell and our protocol, i.e.,

$$E_{sample} < 4/n.$$

**The effect of discretization**  Let $f$ be a $L_f$-Lipschitz and $\forall x, y : f(x,y) \in [0,1]$. This is similar to Bell, i.e.,

$$E_{discr} \leq \frac{L_f^2}{t^2} \tag{19}$$

**The effect of JL theorem**  Let $||A||_{1\to2}$ be the maximum ($\ell_2$)-norm of the columns of $A$. Let $\gamma_2(A) = \min_{A=L^T R} ||L||_{1\to2} ||R||_{1\to2}$. In Ghazi et al. (Ghazi et al., 2024), the authors prove that:

$$\mathbb{E}\left[ (U_{f,C_2^n,\pi} - \tilde{U}_{f,C_2^n,\pi})^2 \right] \leq \left( \frac{\gamma_2(A)}{\epsilon \sqrt{n}} \right)^2.$$

Furthermore, Ghazi et al. (Ghazi et al., 2024, Lemma 17) show that when $f$ is $L_f$-Lipschitz, $\gamma_2(A) \leq O(t \cdot L_f)$. Hence, we have:

$$E_{JL} = O\left( \frac{t^2 L_f^2}{\epsilon^2 n} \right).$$

**The effect of local DP** In Ghazi et al. (Ghazi et al., 2024), the authors prove that

$$\mathbb{E}[(\tilde{U}_{f,C_2^n,\pi} - \tilde{U}_f)^2] \leq \frac{\gamma_2(A)^2}{\epsilon^2 n} + \frac{\gamma_2(A)^2 d}{\epsilon^4 n^2}. \tag{20}$$

Since $f$ is $L_f$-Lipschitz, $\gamma_2(A) \leq O(t \cdot L_f)$ and $d = O((\log t)\epsilon^2 n)$, we have that:

$$E_{DP} \leq O\left(\frac{t^2 L_f^2 \log t}{\epsilon^2 n}\right).$$

In total, the MSE over the sample U-statistic for Ghazi et al. (Ghazi et al., 2024) is:

$$E_{tot} - E_{sample} = E_{discr} + E_{JL} + E_{DP}$$
$$= O\left(\frac{L_f^2}{t^2} + \frac{t^2 L_f^2 \log t}{\epsilon^2 n}\right).$$

### D.1.4. SUMMARY

When assuming $\mathbb{Y} = [0,1]$ and $f$ is $L_f$-Lipschitz, we obtain the following table.

| Protocol | Population MSE | Sample MSE |
|----------|---------------|------------|
| Bell | $O\left(\frac{t^2}{n\epsilon^2} + \frac{L_f^2}{t^2}\right)$ | $O\left(\frac{t^2}{n\epsilon^2} + \frac{L_f^2}{t^2}\right)$ |
| Ghazi | $O\left(\frac{t^2 L_f^2 \log t}{n\epsilon^2} + \frac{L_f^2}{t^2}\right)$ | $O\left(\frac{t^2 L_f^2 \log t}{n\epsilon^2} + \frac{L_f^2}{t^2}\right)$ |
| We | $O\left(\frac{1}{n}\right)$ | $O\left(\frac{1}{|E|} + \frac{1}{n^2\epsilon^2}\right)$ |

*Table 4.* Summary of the MSE for the compared protocols

### D.2. Details on Communication and Computation costs

#### D.2.1. COMMUNICATION COST

Let $\ell$ be the number of bits required to represent one element.

**Bell** In terms of communication cost, the protocol of Bell et al. (Bell et al., 2020) requires to send $O(n\ell t)$ bits in total where $t$ is the number of bins chosen.

**Ghazi** The protocol from Ghazi et al. (Ghazi et al., 2024) allows to reduce the communication cost by requiring that the matrix $A \in \mathbb{R}^{t \times t}$ describing the kernel function is decomposable by two other matrices $L, R \in \mathbb{R}^{d \times t}$ with $d \leq t$. The value $d$ equals $O(\beta^{-2} \log t)$ where $\beta$ is the approximation parameter in the JL theorem. Ghazi et al.'s protocol requires $\beta = 1/\epsilon\sqrt{n}$, which results in $d = O(\epsilon^2 n \log t)$. Hence, Ghazi requires to send $O(\epsilon^2 n^2 \log t\ell)$ bits in total while the shuffled variant GhaziSM requires an additional factor $r$ representing the number servers used for shuffling. In (Balle et al., 2020), the authors show that it suffices to take $r = O\left(\log(n) + \kappa\right)$ for $\epsilon$-DP, where $\kappa$ is the statistical security parameter, yielding $O\left((\log(n) + \kappa)n^2\epsilon^2(\log t)\ell\right)$ bits in communication cost.

**Our protocol** For our protocol Umpc, the communication cost in the online phase is $O(|E|(\ell + C_f^C) + C_\eta^C)$ bits. Under the threat model $\mathcal{M}_{HF}$, the Protocol $\prod_{DLap}$ (Protocol B.1) can be modified to delegate the sharing of noise $\eta$ to a subgroup $Z$. Therefore, the communication cost $C_\eta^C$ is at most $O(n\ell)$. Consequently, Umpc requires a total $O(|E|(\ell + C_f^C) + n\ell)$ bits.

#### D.2.2. PARTY COMPUTATIONAL COST

We measure the computational cost as the number of operations performed by each party.

**Bell**    In Bell, each party $P_i$ only perturbs its one-hot-vector encoded private data $x_i \in \mathbb{R}^t$, which can be done in $O(t)$ operations.

**Ghazi**    The original procotol Ghazi requires each party $P_i$ to perform two matrix-vector products between its private vector data $x_i \in \mathbb{R}^t$ and the matrices $L, R \in \mathbb{R}^{d \times t}$. (Ailon & Chazelle, 2006) gives a construction for Fast JL Transform which allows to perform the product in $O\left(t \log t + \frac{t \log t}{\beta^2}\right)$ operations by enforcing the sparsity of matrices $L, R$ through preconditioning the projection with a randomized Fourier transform. Hence, the Ghazi algorithm requires $O\left(\epsilon^2 n t \log t\right)$ operations. As for GhaziSM, the shuffled model necessitates additional operations to encode each coordinate of the resulting vectors $Lx_i, Rx_i \in \mathbb{R}^d$, yielding $O\left(\epsilon^2 n t \log t + t(\kappa + \log n)\right)$ operations where $\kappa$ is the statistical security parameter.

**Our protocol**    For our protocol Umpc, let $\mathsf{C}_f^T$ denote the computational cost for a single party to evaluate the function $f$ once. Assume we use Protocol $\prod_{\text{DLap}}$ (Protocol B.1). Then, creating distributed Laplace noise requires $O(n)$ overhead per party. With balanced sampling, each party $P_i$ participates in $\lceil 2|E|/n \rceil$ edges where $E$ is the set of edges. Therefore, in Umpc, the computation per party is $O\left(|E| \cdot \mathsf{C}_f^T / n + n\right)$.

### D.2.3. SERVER COMPUTATIONAL COST

The server computational cost is measured as the number of operations the server has to perform in order to compute the U-statistic.

**Bell**    In Bell, the server is required to perform $\binom{n}{2}$ matrix-vector products involving $A \in \mathbb{R}^{t \times t}$, which results in $O(n^2 t^2)$ operations.

**Ghazi**    For the original Ghazi et al.'s protocol (Ghazi), a summation of the vectors sent by the parties is required, resulting in $n \cdot d$ operations. Furthermore, a dot product between vectors of size $d = O(\epsilon^2 n \log t)$ is required. Hence, it requires $O(\epsilon^2 n^2 \log t + \epsilon^2 n \log t) = O\left((\epsilon n)^2 \log t\right)$ operations. For GhaziSM, it requires $O\left((\kappa + \log n)\epsilon^2 n^2 \log t\right)$ operations to the server.

**Our protocol**    For our protocol Umpc, it only requires $n$ additions.

## E. Additional Experiments

In this appendix, we first review the different choices made to implement Umpc. Then, we give additional experimental results.

### E.1. Experimental choices

**Offline phase**    For the experiments, we consider the generation of multiplication triples during the offline phase. There are two types of multiplication triples involved in the protocol:

- 2-party triples for the evaluation of $f$. These triples can be generated using the approach found in (Boyle et al., 2020). Let $\lambda$ be the security parameter, which we set to $\lambda = 128$. This results in $O(\lambda^3 \cdot \log(N_f))$ bits for $N_f$ triples.

- $n$-party triples for the generation of shared noise. To generate these triples, we use the correlated pseudorandomness from (Bombar et al., 2023) which requires $O(\lambda^3 \cdot n^2 \cdot \log(2N_{\text{Noise}}))$ bits to generate $N_{\text{Noise}}$ triples.

**Online phase**    In our experiments, the evaluation of $f$ involves secure comparisons, i.e., compute $[\![x < y]\!]$ given $[\![x]\!]$ and $[\![y]\!]$. For this, we adopt the approach from (Boyle et al., 2019) which employs FSS to enable 2-party secure comparison, resulting in a communication cost of $O(\lambda \ell)$ bits per comparison. For the generation of distributed noise via the Laplacian mechanism, we rely on Protocol B.1, which requires $O(n^2 \ell)$ bits of communication in total.

### E.2. More experimental results

**Datasets**    We consider two additional datasets:

- *Heart Disease dataset* (Andras et al., 1989): This dataset displays a set of features for each patient, e.g., cholesterol, maximum heart rate, etc., along with a target column that describes the diagnosis of heart disease for the patient. The dataset contains 920 entries. After removing the entries with missing values for the columns of interest, the dataset contains 834 entries. We normalize all numerical dataset values.

- *Diabetes dataset* (Strack et al., 2014): The dataset represents ten years (1999-2008) of clinical care at 130 US hospitals and integrated delivery networks. Each row concerns hospital records of patients diagnosed with diabetes, who underwent laboratory, medications, and stayed up to 14 days. The dataset contains 101766 entries. We normalize all numerical dataset values.

**Protocols**  We consider the 4 protocols Ghazi, GhaziSM, Bell and Umpc mentioned in the beginning of Sec 5.

**Gini Mean Difference**  We evaluate the several approaches on the *Diabetes dataset* by computing the Gini Mean Difference (GMD) for the number of distinct generic medications administered during patient encounters. The results are summarized in Table 5. We observe that Umpc results in the lowest MSE while requiring less communication than Ghazi and GhaziSM protocols. Furthermore, although Bell minimizes the communication cost, its inability to complete the execution confirms that its server-side computation cost is a critical bottleneck.

*Table 5.* MSE, total online communication cost and server computation cost of the different protocols for computing the Gini mean difference with $\epsilon = 1$. We run Umpc with $|E| = 2n$ with BalancedSamp, Ghazi, GhaziSM, Bell with $t = 64$. The dataset is taken from (Strack et al., 2014) and contains $n = 101766$ data points. MSE is reported as Mean $\pm$ SEM.

| Protocol | MSE | Comm. cost (bytes) | Server comp. cost (operations) |
|---|---|---|---|
| Bell (Bell et al., 2020) | $> 10h^\dagger$ | $3.26 \times 10^7$ | $2.12 \times 10^{13}$ |
| Ghazi (Ghazi et al., 2024) | $2.2 \times 10^{-2} \pm 2.7 \times 10^{-2}$ | $4.31 \times 10^{11}$ | $8.61 \times 10^{10}$ |
| GhaziSM (Ghazi et al., 2024) | $7.1 \times 10^{-3} \pm 1.5 \times 10^{-2}$ | $5.90 \times 10^{13}$ | $1.18 \times 10^{13}$ |
| Umpc (Protocol 3.1) | $1.8 \times 10^{-8} \pm 3.0 \times 10^{-5}$ | $1.35 \times 10^8$ | $1.02 \times 10^5$ |

$^\dagger$ The baseline exceeded the maximum allocated cluster runtime limit of 10 hours for a single run, at which point the execution was terminated.

**Kendall's $\tau$ coefficient**  Let $x_i = (y_i, z_i)$ where $y_i, z_i \in [0, 1]$ represent respectively the resting blood pressure and the serum cholesteral for party $i \in [n]$. The goal is to compare the several metrics mentioned in Sec 5 when computing Kendall's $\tau$ coefficient for data points $\{x_i\}_{i \in [n]}$ for $\epsilon = 1$.

Figure 5 shows the total communication cost, the MSE, the computation cost per party and the computation cost of the server for the different protocols when computing the Kendall's $\tau$ coefficient. For each point on the graph, data points are sampled uniformly without replacement from the dataset. While Umpc has a higher communication cost than Bell, it achieves lower MSE and server computation cost.

**Duplicate Pair Ratio**  We focus on computing $U_f$ where $f(x, y) = \mathbb{I}[x = y]$ such that $x, y$ are categorical data points. We use the same dataset as the previous example, i.e., the Heart Disease dataset (Andras et al., 1989) to obtain a diagnosis of heart disease for each patient. In this context, the input space is $\mathbb{X} = \{0, \dots, 4\}$.

Figure 6 presents the total online communication cost, the MSE, the per-party and server computation costs for the duplicate pair ratio. Although Bell is optimal in both total communication cost and per-party computation cost, it transfers the main computational load to the server. One can observe a clear distinction for the MSE between achieving central-DP — Umpc, GhaziSM — and LDP, i.e., Bell and Ghazi.

## F. Balanced sampling

A number of properties of our protocol depend on the maximal degree $\delta_G^{max}$ of the hypergraph $G = (V, E)$. The best is to have a small $\delta_G^{max}$ and hence to sample a hypergraph $G$ which is a regular as possible, i.e., every instance $x_i$ is involved in about the same number of edges $e \in E$ (one can't avoid a difference of 1 in the degrees of the vertices as $k|E|$ may no be a multiple of $n$). The algorithm below is a simple way to generate such hypergraph $G$.

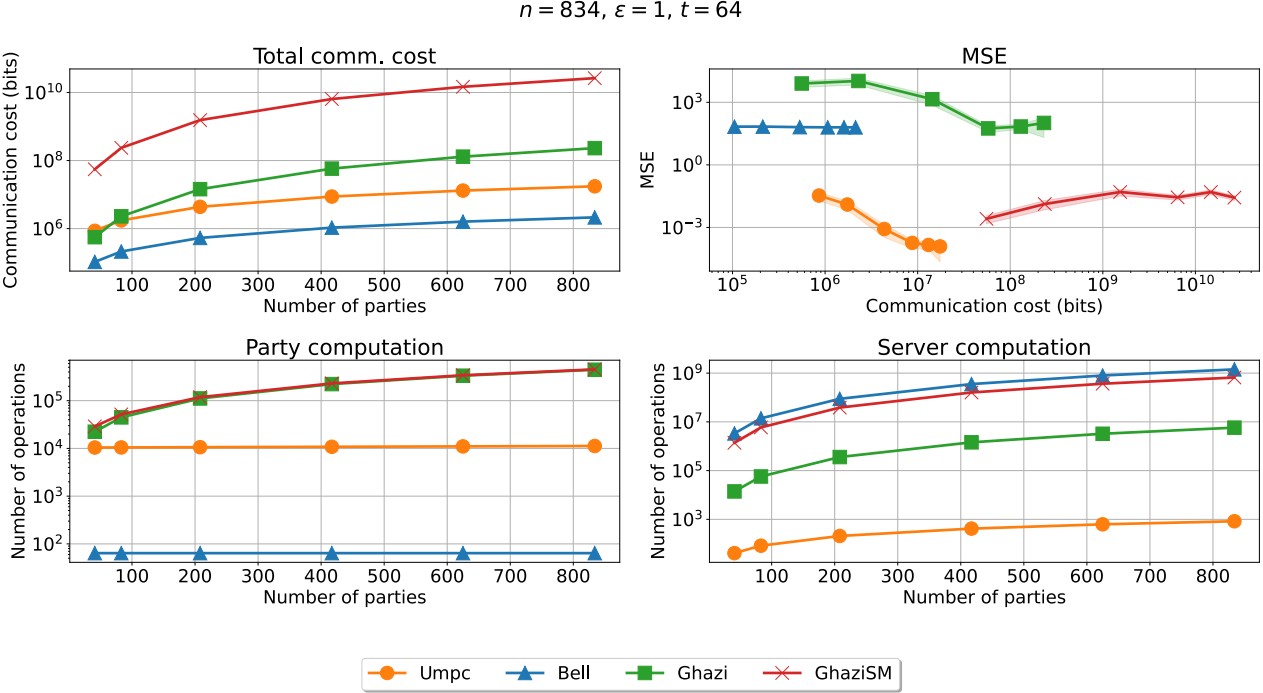

*Figure 5.* Online total communication cost, MSE, per-party and server computation costs comparison for computing the Kendall's $\tau$ coefficient with protocols Bell, Ghazi, GhaziSM and Umpc for $\epsilon = 1$ and $t = 64$. The dataset is taken from (Andras et al., 1989). For Umpc, we sample $2n$ edgesfor $|E|$.

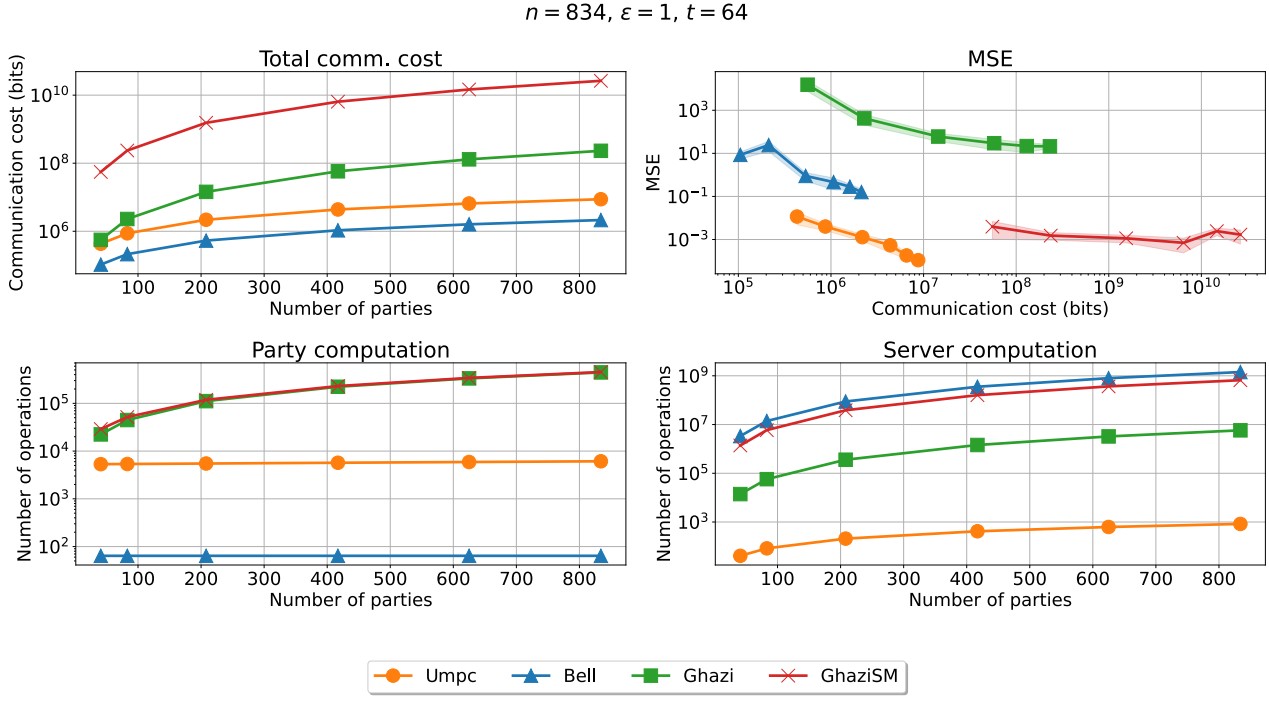

*Figure 6.* Online total communication cost, MSE, per-party and server computation costs comparison for computing $U_f$ with $f(x, y) = \mathbb{I}[x = y]$ with $\epsilon = 1$ and $t = 64$. Data represents the diagnosis of heart diseases of patients from (Andras et al., 1989). For Umpc, we sample $2n$ edges for $|E|$.

---
**Algorithm 1** BalancedSamp$(m, k, n)$

---
**Input:** Size of the set of edges $m$, kernel degree $k$, number of parties $n$
 1: Set $E = \{\}$.
 2: Set $M = \left\lceil \frac{km}{n} \right\rceil$.
 3: Set $p_i = M$ for $i \in [n]$.
 4: **while** $|E| < m$ **do**
 5:     **if** $p$ contains less than $k$ non-zero values **then**
 6:         **restart**
 7:     **end if**
 8:     Sample without replacement $v_1, \ldots, v_k$ from $[n]$ where the probability to select $v_i$ is proportional to $p$.
 9:     Add the edge $(v_1, \ldots, v_k)$ to $E$.
10:     **for** $i \in [k]$ **do**
11:         **if** $p_{v_i} > 0$ **then**
12:             $p_{v_i} \leftarrow p_{v_i} - 1$
13:         **end if**
14:     **end for**
15: **end while**
16: Return $E$.

---

