# OpenReview forum: "Accurate, Private, Secure, Federated U-statistics with Higher Degree"
_ICML.cc/2026/Conference — ICML 2026 regular_

### Official Review · Reviewer_npNJ · 2026-02-23

**Soundness:** 3
**Presentation:** 3
**Significance:** 3
**Originality:** 3
**Overall Recommendation:** 4
**Confidence:** 4

**Summary:**

This paper investigates the problem of securely computing U-statistics of degree $k \ge 2$ in a federated setting under central differential privacy. The authors propose a protocol that integrates hypergraph-based sampling, secure multi-party computation for kernel evaluation, and distributed Gaussian noise generation to achieve $(\epsilon,\delta)$-DP guarantees. The paper presents a detailed analysis of communication and round complexity, and empirically compares the proposed method against local DP, shuffled DP, and MPC-based baselines for degree-2 U-statistics, including the Gini mean difference and Kendall’s $\tau$. Experimental results indicate that the proposed approach achieves improved accuracy under central DP while maintaining competitive communication costs.

**Compliance With Llm Reviewing Policy:**

Affirmed.

**Final Justification:**

After considering the authors’ response during the rebuttal and discussion phase, I appreciate the detailed clarifications provided. In particular, the explanation of maintaining $|E| = O(n)$ as a practical trade-off between communication cost and estimation error, as well as the additional example for $k>2$, help improve the understanding of the proposed approach. Several of my earlier concerns, such as the baseline positioning and the sampling inconsistency, have been adequately addressed.
However, some limitations remain. The empirical evaluation is still primarily focused on $k=2$, and the evidence for higher-degree scalability relies on a limited illustrative example rather than a systematic evaluation. While the rebuttal provides useful intuition, these aspects would require additional experimental validation and are unlikely to be fully resolved within the rebuttal phase.

I find the work to be technically sound and addressing a meaningful problem. The rebuttal improves the clarity of the paper but does not fundamentally change my assessment. Therefore, I maintain my original score.

**Key Questions For Authors:**

1. The communication complexity scales with $|E|$, which may grow as $O(n^k)$. Could the authors clarify practical regimes of $k$ and $|E|$ under which the protocol remains scalable in realistic settings?

2. Since baselines operate under different privacy models, could the authors discuss the fairness of cross-model comparisons and how a trusted-aggregator central DP baseline would perform as a reference point?

3. The experiments focus on $k=2$. Could the authors comment on empirical or practical scalability for $k>2$, or provide insights into how performance is expected to change?

4. Could the authors clarify the discrepancy in sampling sizes (e.g., $2n$ vs. $0.02n^2$) across Figures 1 and 2, and explain the rationale behind these choices?

**Limitations:**

Yes

**Strengths And Weaknesses:**

Strengths

- Soundness. The proposed protocol is modular and well-structured, with a clear separation between secure kernel evaluation, noise generation, and aggregation, which facilitates both analysis and implementation. The communication complexity analysis is carried out phase-by-phase and appears internally consistent. The technical development is coherent, and no obvious correctness issues were identified, suggesting that the approach is technically sound under the stated assumptions.

- Presentation. The paper is generally clearly written and logically organized. The protocol description, theoretical analysis, and empirical evaluation are well separated, which improves readability and makes it easier to follow the overall workflow of the method. Key components of the approach are introduced in a structured manner.

- Significance. Computing U-statistics of degree $k \ge 2$ under federated constraints with central differential privacy is a meaningful and relatively underexplored problem, particularly in privacy-sensitive applications where secure aggregation alone is insufficient. The ability to compute higher-order statistics extends beyond standard first-order aggregation commonly used in federated learning.

- Originality. While the individual building blocks (MPC, DP, sampling) are known, their integration via hypergraph-based sampling and distributed noise generation for higher-degree U-statistics is non-trivial and well motivated. The design reflects a thoughtful combination of techniques tailored to this problem setting.

Weaknesses

- Scalability discussion. The communication complexity scales linearly with $|E|$, which may grow as $O(n^k)$. Although experiments use moderate $|E|$ for $k=2$, this raises concerns about practical scalability for higher-degree settings, which are part of the paper’s main claim. A clearer discussion of realistic parameter regimes would strengthen the paper.

- Baseline positioning. Since the compared baselines operate under different privacy models (local DP, shuffled DP, central DP), the comparison is not entirely apples-to-apples. A clearer discussion of fairness and contextual positioning, including the role of trust assumptions, would help better interpret the empirical results.

- Experimental scope. All empirical results focus on $k=2$. While this is a natural starting point, it limits empirical support for the broader claim of handling $k \ge 2$. Additional experiments or a more explicit discussion of expected behavior for higher-degree statistics would improve the completeness of the evaluation.

- Experimental rigor. Some presentation details could be improved, such as clarifying sampling size choices across figures (e.g., $2n$ vs. $0.02n^2$ in Figures 1 and 2) and reporting error bars for MSE. Since the method involves stochastic sampling, providing variance information would strengthen the empirical validation and improve interpretability of the results.

- Reproducibility / anonymization. The authors may wish to double-check that the provided code repository complies with the conference’s double-blind requirements, as external links may inadvertently reveal identifying information.

---

> ### Author Rebuttal · Authors · 2026-03-31
>
> We thank the reviewer for their supportive and insightful assessment of our work.  Below, you'll find our responses, which we hope clarify the points you raised.
>
> > **1. The communication complexity scales with $|E|$, which may grow as $O(n^k)$. Could the authors clarify practical regimes $k$ of |E| and under which the protocol remains scalable in realistic settings?**
>
> We agree with the reviewer that $|E|$ may grow as $O(n^k)$. Still, in practice sampling more than $|E|=O(n)$ edges is not very useful, in the sense that it increases the cost and can't significantly further improve the total error. In particular, it is unavoidable that $E_{sample} = O(1/n)$.  $E_{inc}$ is at best $O(1/|E|)$, an $|E|$ larger than $O(n)$ can't result in a total error with order of magnitude smaller than $O(1/n)$.  Hence, $|E|=O(n)$ will be the most reasonable trade-off from a practical point of view, which is the reason we keep $|E|/n^{k-1}$ constant.
> The constant in $E_{sample}=O(1/n)$ depends on the distribution of $f(e)$, and hence constants $|E|/n^{k-1}$ of smaller order of magnitude won't improve the error much. We will make this dependency more explicit in the text.
>
> ---
>
> > **2. Since baselines operate under different privacy models, could the authors discuss the fairness of cross-model comparisons and how a trusted-aggregator central DP baseline would perform as a reference point?**
>
> We agree with the reviewer that the comparison is challenging when trust assumptions vary. To ensure a fair comparison, let us explain the different models:
>   - The Local DP setting requires the least trust (the aggregator is untrusted). This provides the strongest privacy but suffers from a high error rate of $O(1 / n)$.
>   - Central DP assumes a trusted aggregator who receives raw data of the parties and adds appropriate DP noise, achieving an $O(1 / n^2) error rate.
>   - Shuffled DP is an intermediate model that improves upon LDP utility but often relies on the assumption of trusted shufflers to anonymize the inputs when send to the untrusted aggregator.
>
> Our protocol is designed to achieve utility comparable to central DP without requiring a trusted aggregator. By using MPC, we provide a "Distributed-Trust" model where: we match the error rate of central DP, the trust model is distributed across parties.
> We will update the paper to make this comparison clear to the reader.
>
> ---
>
> > **3. The experiments focus on $k=2$. Could the authors comment on empirical or practical scalability for $k>2$, or provide insights into how performance is expected to change?**
>
> While our benchmark focuses on $k=2$ (mainly because that is the only setting in which there are baselines in the literature to which we can compare), our protocol is designed to support $k > 2$ with linear communication cost by maintaining $|E|=O(n)$ (as argued in Q1). To demonstrate this, we will include a use case on triangle counting in the final version.  Triangle counting is the problem where, given a graph $G=(W,F)$, one wants to know the number of triangles, i.e., the number $T_G$ of unordered triples $(x_1,x_2,x_3)$ such that $(x_1, x_2),(x_2,x_3),(x_1,x_3)\in F$.
> $T_G$ can be obtained as a U-statistic with kernel of degree $k=3$, i.e., $T_G=\sum_{(x_1,x_2,x_3)\in\binom{[W]}{3}}\mathbb{I}[ (x_1,x_2)\in F ] \cdot \mathbb{I}[(x_2, x_3)\in F] \cdot \mathbb{I}[(x_1, x_3) \in F]$.
> This requires every data owner $w\in W$ to store the list of vertices $w'\in W$ which are adjacent to $w$, i.e., $(w, w') \in F$. Using the dataset [1] (1612010 triangles, |W| = 4039), we obtain the following results for $\epsilon = 1$:
>
> |Size E|RMSE$/T_G$|std. dev.|Comm. cost per party|
> |:---|:---|:---|:---|
> |$10^5$| $0.25574$ |$0.19712$ |$48$ KB|
> |$10^6$| $0.07022$ |$0.05397$ |$470.4$ KB|
> |$10^7$| $0.02375$ |$0.01951$ |$4.8$ MB|
>
> These results validate our framework's ability to handle scalability in large-scale networks without the combinatorial explosion of U-statistics. Nevertheless, please note that several special-purpose triangle counting algorithms have been proposed in the literature.  Our general-purpose algorithm has no ambition to compete with those.  The above is only aimed at a simple illustration of our approach for $k=3$.
>
> ---
>
> > **4. Could the authors clarify the discrepancy in sampling sizes (e.g., $2n$ vs. $0.02 n^2$ ) across Figures 1 and 2, and explain the rationale behind these choices?**
>
> We thank the reviewer for identifying this error in the manuscript. We apologize for the confusion. To clarify, all empirical results reported in the paper were obtained using $|E| = 2n$. This regime allows to keep the communication linear while having reasonable utility. We will correct the incorrect captions when releasing the final version.
>
> [1] McAuley J.,  Leskovec J, Learning to Discover Social Circles in Ego Networks. NIPS, 2012. link: https://snap.stanford.edu/data/ego-Facebook.html

---

> > ### Author Rebuttal · Reviewer_npNJ · 2026-04-03
> >
> > Thank you for the detailed rebuttal and the additional clarification on maintaining $|E| = O(n)$ as a practical trade-off. The discussion, along with the triangle counting example, helps improve the understanding of scalability for $k > 2$.
> >
> > However, some limitations remain. The empirical evaluation is still primarily focused on $k=2$, and the evidence for higher-degree scalability relies on a single illustrative example rather than a systematic evaluation. As a result, while the rebuttal improves the clarity of the method, it does not fully address the limitations in experimental scope.
> > As a follow-up, could the authors further clarify whether the observed trade-off $|E| = O(n)$ consistently provides stable accuracy across different types of U-statistics (beyond the triangle counting example), or whether its effectiveness is problem-dependent?

---

> > > ### Author Response · Authors · 2026-04-06
> > >
> > > We appreciate the reviewer’s follow-up and the opportunity to further clarify the scalability for $k>2$.
> > >
> > > > ... The empirical evaluation is still primarily focused on k=2,  and the evidence for higher-degree scalability ....  it does not fully address the limitations in experimental scope.
> > >
> > > Our paper focuses on experiments with $k=2$ as this is the only setting where we can compare to existing methods in the SOTA. For $k>2$, we outperform the SOTA because there are no existing generic algorithms for this task. The scalability for $k>2$ is not necessarily something which requires experimental evaluation.  Independently of $k$, our algorithm averages over $|E|=O(n)$ evaluations of the kernel $f$, i.e., the dominant cost is $n CC_f$ where $CC_f$ is the cost of a single evaluation of the kernel $f$. We observe indeed experimentally that the cost is always linear in $n$.
> > >
> > > If the reviewer asks for the scalability in $k$,  one can only answer that this is very problem-dependent and not really related to our specific contribution.
> > > * Consider that in U-statistics $k$ is typically not a large number.  It may be $1,2,3,4$ ... but is not supposed to scale to large values (there are no popular practical use cases with large $k$).
> > > * If one insists on the theoretical question of what happens for large $k$, then the answer is likely that whatever is the value of $k$, there exist very hard problems.  Even if $k=1$, one could find kernel functions $f$ which are NP-hard in the length (in bits) of the single input $x_1$ of $f(x_1)$ or worse (possibly undecidable).  Similarly, for increasing $k$ one can find functions $f(x_1 ... x_k)$ which have a cost exponential in $k$.  Still, for most practical problems $f$ is not a complex function and even for larger $k$, evaluating the kernel function $f$ is cheap.  Therefore, $CC_f$ depends much more on the definition of the kernel function $f$ used than on the specific value of $k$.
> > > * In the exceptional case where $f$ is hard to compute, one could (as the Bell and Ghazi baselines) discretize values in $t$ bins, reducing $f$ to a lookup in a pre-computed $t^k$-size tensor (if that fits in storage space).
> > > * Moreover, please consider that even if $f$ is cheap to compute, we (nor any other privacy-preserving algorithm for U-statistics in the literature) don't have the ambition to solve NP-hard problems in polynomial time.  Suppose you want to count $k$-cliques in a graph for increasing $k$ (ignoring the fact that usually the word "U-statistic" does not refer to the graph setting and that usually U-statistics have a rather small degree).  Given a $k$-tuple, checking whether it represents a clique costs $O(k^2)$.  We will be able to approximate the number of $k$-cliques well in time $O(nk^2)$.  Still, this remains consistent with the fact that the problem of finding a $k$-clique remains NP-hard: one would need to iterate over all $n^k$ tuples to be sure whether there is at least one $k$-clique, but even then the differential privacy noise would make it impossible to get this knowledge from the output.
> > >
> > > In conclusion, the scalability in $k$ is not a very relevant question: the key question is the complexity of the kernel function $f$ which can be (usually) low but (exceptionally) also very high depending on the problem.
> > >
> > > > could the authors further clarify whether the observed trade-off $|E| = O(n)$ consistently provides stable accuracy across different types of U-statistics (beyond the triangle counting example), or whether its effectiveness is problem-dependent?
> > >
> > > As said, because $E_{sample}=O(1/n)$, there is no hope to get the overall result better than $O(1/n)$.  Even so, it can become a bit better by choosing $|E|$ higher.  It is important to note that if $|E|>n$ the values  $\{f(e)\}\_{e\in E}$ to average are not independent, e.g., $f(x_1,x_2)$ and $f(x_1,x_3)$ are not independent as they both depend on $x_1$.  This ensures that even if we would choose $|E|=O(n^2)$ we would have no guarantee that $E_{inc}$ is better than $O(1/n)$.  This effect has been studied in depth in probability theory literature, e.g., in [1].  It is not our ambition to improve on that SOTA or to experimentally confirm it again.  Our paper aims at making a rather orthogonal algorithmic contribution in the privacy-preserving federated setting.
> > >
> > > Based on the existing literature (and our experimental results consistent with that literature) we can only say that,
> > > * yes, $|E|=O(n)$ gives consistently a stable accuracy with $O(1/n)$ MSE across different types of U-statistics.
> > > * whether it is possible to achieve slightly better (but still $O(1/n)$) MSE by further increasing $|E|$ to a value between $n$ and $n^k$ is problem-dependent (due to the fact that $f(x_1,x_2)$ and $f(x_1,x_3)$ are not independent) and is a non-trivial problem in probability theory which is investigated also in the non-private non-federated setting.
> > >
> > > [1] Blom, G. (1976). Some properties of incomplete U-statistics. Biometrika, 573-580.

---

### Official Review · Reviewer_nSCW · 2026-03-10

**Soundness:** 3
**Presentation:** 2
**Significance:** 3
**Originality:** 2
**Overall Recommendation:** 4
**Confidence:** 2

**Summary:**

Addressing the challenge of privacy-preserving computation for U-statistics in distributed environments, the authors propose a generalised protocol, U-MPC, based on secure multi-party computation (MPC). The effectiveness of this approach is demonstrated through experimental validation.

**Compliance With Llm Reviewing Policy:**

Affirmed.

**Final Justification:**

The author has addressed my concerns.

**Key Questions For Authors:**

1) Baseline comparisons are limited, with only LDP methods evaluated in the paper; additional MPC-related comparison methods are required
2) Relatively small dataset size; lacks analysis of the impact of the |E|/n ratio on performance
3) The scope of the paper's innovation needs clarification

**Limitations:**

yes

**Strengths And Weaknesses:**

Strengths:
1) Sound theoretical foundation with high credibility. The paper provides a detailed analysis of the protocol’s security, communication complexity and MSE, and systematically compares it with existing work.
2) Comprehensive and effective experimental validation (Communication costs, MSE, computational costs). The results demonstrate the advantages of this method.
3) Well-extended methodology supporting U-statistics of arbitrary order.

Weaknesses:
1) Baseline comparisons are limited, with only LDP methods evaluated in the paper; additional MPC-related comparison methods are required
2) Relatively small dataset size; lacks analysis of the impact of the |E|/n ratio on performance
3) The scope of the paper's innovation needs clarification. The text should more clearly define the innovation of this paper as "the first general-purpose MPC protocol for U-statistics that supports arbitrary orders and implements central dynamic programming", and emphasise the system optimisations in areas such as sampling strategies and noise generation. Avoid using phrases such as ‘novel protocol’ that may lead to overinterpretation.

---

> ### Author Rebuttal · Authors · 2026-03-31
>
> We thank for the insightful feedback. We believe that their suggestions will strenghten the paper's completeness.  Below, you'll find our responses, which we hope clarify the points you raised.
>
> > **1. Baseline comparisons are limited, with only LDP methods evaluated in the paper; additional MPC-related comparison methods are required**
>
> To the best of our knowledge, the papers mentioned in the "Related works" section are the only articles that deal with the same setting as ours, i.e., the federated setting. As for MPC-related comparison methods, we acknowledge that other approaches within the MPC framework could be considered, e.g., Fully Homomorphic Encryption (FHE) instead of Additive Secret Sharing (AddSS). We chose AddSS for three strategic reasons. First, FHE introduces a computational overhead that is orders of magnitude higher than AddSS. Second, the injection of distributed DP noise is done almost for "free" with AddSS.  Third, for computing individual applications of the kernel function, we use a "local" additive secret sharing, which generalizes into a secret sharing over all parties "for free" in our specific setup; this doesn't seem to work as easily for Shamir secret sharing or FHE.
> We will update the "Related works" section and add a discussion regarding the trade-offs between different MPC primitives in the context of our problem to motivate our choice.
>
> ---
>
> > **2. Relatively small dataset size;**
>
> To answer the experimental questions there was no strong need to consider very large datasets.  The scaling to large amounts of data is shown to a large extent by our theoretical analysis, which shows that computation/communication cost does not significantly depend on the size of the dataset, but much more on the desired precision of the result.  This is a typical property for statistics which can be approximated by sampling.
> Nevertheless, we are happy to illustrate this better, we will include in the final version an experiment on the Diabetes dataset [1] ($n = 101766$ entries) to analyze the Gini Mean Difference of hospital stay durations. The results for this dataset are similar to the results presented already and hence confirm our theoretical results. With $|E| = 5n$ and $\epsilon = 1$, we achieved an average MSE of $1.03 \times 10^{-8}$  (std. dev. $1.44 \times 10^{-8}$), matching the theoretical results while having a communication cost of 3.3 KB per party.
>
> ---
>
> > **lacks analysis of the impact of the |E|/n ratio on performance**
>
> It is natural to fix $|E|/n^{k-1}$, or for arbitrary $k$, $|E|/n^{k - 1}$. The reason is as follows: increasing $|E|$ decreases the error $E_{inc}$ but also increases the cost.
> It doesn't make practical sense to optimize $E_{inc}$ if the other costs $E_{sample}$ and/or $E_{DP}$ have a higher order of magnitude, hence as $E_{sample}=O(1/n)$, it doesn't help much to increase $|E|$ above $O(n)$. The constant in $E_{sample}=O(1/n)$ depends on the distribution of $f(e)$.  Choosing a larger $|E|/n^{k-1}$ can't significantly improve the order of magnitude of the total error.
> We will make this more explicit in the text.
>
> ---
>
> > **3. The scope of the paper's innovation needs clarification. The text should more clearly define the innovation of this paper as "the first general-purpose MPC protocol for U-statistics that supports arbitrary orders and implements central dynamic programming", and emphasise the system optimisations in areas such as sampling strategies and noise generation. Avoid using phrases such as ‘novel protocol’ that may lead to overinterpretation.**
>
> First, we would like to respectfully point out that our paper concerns differential privacy (DP), we don't mention "dynamic programming" in our paper.   Also, we don't define a concept "orders", our problem parameters are only "number of parties/instances" and "kernel degree".
> We believe we do specify the innovation of our paper in Section 1.2. Most important is our first claim, i.e., "We provide a generic protocol for this task that scales well in the kernel degree k, the number of parties n and the size of the discretization". In other words, we don't present the first protocol for this problem but we present the first protocol which -- in contrast to the state of the art -- achieves the mentioned scalability properties. We will make the sentence more specific by saying "We provide the first" rather than "We provide a".
>
>
> [1] Strack B., DeShazo J., Gennings C., Olmo J., Ventura S., Cios K., Clore J., “Impact of HbA1c Measurement on Hospital Readmission Rates: Analysis of 70,000 Clinical Database Patient Records,” BioMed Research International, vol. 2014, Article ID 781670, 11 pages, 2014. link: https://archive.ics.uci.edu/dataset/296/diabetes+130-us+hospitals+for+years+1999-2008

---

> > ### Author Rebuttal · Reviewer_nSCW · 2026-04-04
> >
> > Thank you for your reply, and I will improve my rating.

---

### Official Review · Reviewer_PAAB · 2026-03-13

**Soundness:** 3
**Presentation:** 4
**Significance:** 3
**Originality:** 4
**Overall Recommendation:** 4
**Confidence:** 3

**Summary:**

This paper provides an algorithm to compute U-statistics with central differential privacy in a federated manner -- this is achieved by using a multiparty algorithm together with MPC. Theoretical performance guarantees together with some experiments are provided, and the algorithm appears to work well in that it has the lowest MSE for comparable communication cost out of several baselines.

**Compliance With Llm Reviewing Policy:**

Affirmed.

**Final Justification:**

This is a solid technical paper -- I am highly supportive of acceptance.

**Key Questions For Authors:**

see above

**Limitations:**

yes

**Strengths And Weaknesses:**

Strengths:

1. The problem is clean, and the algorithm proposed achieves some improvement over the baselines considered.
2. The paper is quite well written and the work is well executed.
3. The algorithmic idea is quite novel and original

Weaknesses:

1. The paper is missing comparisons with some related work -- for example:
"On Differentially Private U-Statistics", Chaudhuri, Loh, Pandey and Sarkar, NeuRIPS 2024.

2. There is also connections between DP U statistics and DP graphs literature -- for example, many graph statistics such as counting the number of triangles in a graph are essentially U statistics problems. The paper could benefit from a discussion of this connection -- in terms of what it enables.

---

> ### Author Rebuttal · Authors · 2026-03-31
>
> We thank the reviewer for their positive and thorough feedback. Below, you'll find our responses, which we hope clarify the points you raised.
>
> > **1. The paper is missing comparisons with some related work -- for example: "On Differentially Private U-Statistics", Chaudhuri, Loh, Pandey and Sarkar, NeuRIPS 2024.***
>
> While the cited paper addresses U-statistics under differential privacy, our contribution is distinct in two points.  First, "On Differentially Private U-Statistics", Chaudhuri, Loh, Pandey and Sarkar, NeuRIPS 2024 operates in a central rather than a federated setting, and the trusted aggregator has access to all sensitive data belonging to possibly distinct data owners.  In our setting, this is assumed to be unacceptable due to privacy concerns. It seems non-trivial to generalize this work to a federated setting.
> The second distinction is that the paper cited relies on the assumption that the kernel $h(X_1, \ldots, X_k)$ is sub-Gaussian while our protocol does not make such assumptions on the distribution of the kernel function.
> We will update our "Related works" section to stress more that we here consider a federated setting and to contrast the different trust and security assumptions in the federated and centralized settings.
>
> ---
>
> >  **2. There is also connections between DP U statistics and DP graphs literature -- for example, many graph statistics such as counting the number of triangles in a graph are essentially U statistics problems. The paper could benefit from a discussion of this connection -- in terms of what it enables.**
>
> We appreciate the reviewer's suggestion to connect our work to the DP graph literature. Here too, we will, in our "Related works" section, point the reader to this related field.  Indeed, triangle counting is an instance of counting results on conjunctive queries, a topic on which several papers have been published, e.g., [1], [2]. While the two problems (U-statistics and graph statistics) are not equivalent (when instances become vertices, the size to represent their properties/adjacencies is not constant anymore), the similarity of the problems may interest the reader.
>
>
> [1] Narayan, A., & Haeberlen, A. (2012). {DJoin}: Differentially private join queries over distributed databases. In 10th USENIX Symposium on Operating Systems Design and Implementation (OSDI 12) (pp. 149-162).
>
> [2] Dong, W., & Yi, K. (2022, June). A nearly instance-optimal differentially private mechanism for conjunctive queries. In Proceedings of the 41st ACM SIGMOD-SIGACT-SIGAI Symposium on Principles of Database Systems (pp. 213-225).

---

> > ### Author Rebuttal · Reviewer_PAAB · 2026-04-03
> >
> > Thank you, please add this discussion to the main body of the paper.

---

### Official Review · Reviewer_chtm · 2026-03-13

**Soundness:** 3
**Presentation:** 3
**Significance:** 3
**Originality:** 2
**Overall Recommendation:** 4
**Confidence:** 2

**Summary:**

The paper studies computing U-statistic in a differentially private way in federated setting using MPC. U-statistic of degree $k \geq 2$ depends on interactions across multiple parties and the problem requires revealing only a differentially private estimate to the aggregator without revealing input or computation across parties. The approach involves sampling $k$-tuples (treating it as a hypergraph), using secret sharing for computing the kernel and then privatizing. Experiments show their approach obtains smaller error with comparable communication and computation.

**Compliance With Llm Reviewing Policy:**

Affirmed.

**Final Justification:**

The work has an overall good set of technical contributions and concerns about empirical evaluations, and contributions were addressed through rebuttal. While I maintain my positive evaluation of the work, I do not have strong enough opinions to champion this work.

**Key Questions For Authors:**

In the experimental section, why is the reported computation cost from the bounds and not actual runtime?

**Limitations:**

Yes.

**Strengths And Weaknesses:**

The paper is well written and clearly motivated. Their approach of approximating with sampling, secure computation and then noisy computation helps separate errors as well, which might be useful for implementation. The proposed sampling technique for sampling $k$-tuples helps control the number of tuples one party is involved in, which in turn controls the sensitivity. The overall approach helps avoid dependencies on the discretization parameter $t$. The experiments are limited to $k=2$ and comparison of preprocessing cost to baselines is a bit unclear.

---

> ### Author Rebuttal · Authors · 2026-03-31
>
> Thank you for your insightful feedback and careful reading. Below, you'll find our responses, which we hope clarify the points you raised.
>
>
>
> > **1. In the experimental section, why is the reported computation cost from the bounds and not actual runtime?**
>
>  While empirical runtimes are valuable, empirical runtimes are influenced by a lot of hard to control external factors such as the current load of the network, the network latency, the load of the processor, the operating system scheduling strategy, etc.  These are typical issues when evaluating distributed algorithms.  We instead use as a metric the number of operations performed by the algorithms, which only measures the costs inherent to the algorithms.  This metric is sufficient to answer the experimental questions, which ask for the comparison with state of the art strategies (rather than the orthogonal question of how the efficiency of distributed algorithms depends on environment parameters such as OS / network latency).  It is more stable and reproducible, and for a fixed environment it is asymptotically equivalent to the empirical costs.
> In the final version, we will motivate this methodological choice in Section 6.1.
>
> ---
>
>
> > **2. The experiments are limited to $k=2$ and comparison of preprocessing cost to baselines is a bit unclear.**
>
>  We agree with the reviewer that we did not explicitely compare the cost of preprocessing in the paper. In general, when considering the scaling of algorithms to larger volumes of operations, the preprocessing costs are negligible in comparison to the total costs.  That is why existing literature often doesn't discuss it in depth.  Our paper provides more details than the articles describing the baselines we compare to. Essentially, our method based on MPC doesn't have data-independent preprocessing costs.  Sometimes, the generation of beaver triples is considered as preprocessing, but its cost is sublinear in |E| and can hence be amortized over the actual computations of the algorithm.  In contrast, the existing methods (Bell and Ghazi) we compare to, need to construct or broadcast a $t\times t$ matrix, which scales poorly in the size of the discretization.  E.g., if one wants 4 or 5 digits of precision on the input data one needs a matrix of size $10^4 \times 10^4 = 10^8$ respectively $10^{10}$ already. We can conclude that especially for non-trivial discretization sizes the preprocessing cost of our algorithm is much smaller than those of the baselines Bell and Ghazi.
>
>  We presented experiments with $k=2$ because this is the only setting where we can compare our algorithm to existing baselines, to the best of our knowledge, no earlier article proposed a generic algorithm for private computation of U-statistics with kernel of degree $k > 2$.  We will add in the final version an experiment where we write the triangle counting problem in a graph as a U-statistic (see answer to reviewer npNJ for full details). The results are in line with the theory developed in our paper.

---

> > ### Author Rebuttal · Reviewer_chtm · 2026-04-03
> >
> > I thank the authors for their response. Going through the rest of reviewrs concerns and the replies, I maintain my score.

---

### Decision · Program_Chairs · 2026-04-30

**Decision:**

Accept (regular)

**Comment:**

The authors study the problem of securely computing U statistics in distributed setting protecting the data owned by a user while cooperating on computing the U statistic. U  statistics with k degrees of freedom are a generalization of many well-known statistics (mean is a k=1 U statistic, avg pairwise distance is a k=2 U statistic, etc). The authors present novel and improved results on computing U Statistics in MPC. The method is based on sampling k tuple using secret sharing and noise addition for privacy. The theoretical analysis shows improved results over prior work. The authors also perform some empirical analysis. Overall the reviewers unanimously found the paper of interest. Some reviewers suggested to include other works in the related work and baselines. Moreover, some reviewers found the interest from the ICML community to be limited to a smaller set of attendees. I suggest the authors to use the reviews to improve their work for the camera ready.